# Anti-platelet factor 4/polyanion antibodies mediate a new mechanism of autoimmunity

Thi-Huong Nguyen[1,2], Nikolay Medvedev[2], Mihaela Delcea[1,2] & Andreas Greinacher[1]

Antibodies recognizing complexes of the chemokine platelet factor 4 (PF4/CXCL4) and polyanions (P) opsonize PF4-coated bacteria hereby mediating bacterial host defense. A subset of these antibodies may activate platelets after binding to PF4/heparin complexes, causing the prothrombotic adverse drug reaction heparin-induced thrombocytopenia (HIT). In autoimmune-HIT, anti-PF4/P-antibodies activate platelets in the absence of heparin. Here we show that antibodies with binding forces of approximately 60–100 pN activate platelets in the presence of polyanions, while a subset of antibodies from autoimmune-HIT patients with binding forces ≥100 pN binds to PF4 alone in the absence of polyanions. These antibodies with high binding forces cluster PF4-molecules forming antigenic complexes which allow binding of polyanion-dependent anti-PF4/P-antibodies. The resulting immuno-complexes induce massive platelet activation in the absence of heparin. Antibody-mediated changes in endogenous proteins that trigger binding of otherwise non-pathogenic (or cofactor-dependent) antibodies may also be relevant in other antibody-mediated autoimmune disorders.

[1] Institute for Immunology and Transfusion Medicine, University Medicine Greifswald, Greifswald 17475, Germany. [2] ZIK HIKE, Center for Innovation Competence, Humoral Immune Reactions in Cardiovascular Diseases, University of Greifswald, Greifswald 17489, Germany. Correspondence and requests for materials should be addressed to T.H.N. (email: nguyent@uni-greifswald.de) or to A.G. (email: greinach@uni-greifswald.de).

The immune response to complexes of the positively charged chemokine platelet factor 4 (PF4, CXCL4) and polyanions[1], shows a spectrum of symptoms in patients with asymptomatic serum positivity for anti-PF4/polyanion antibodies (anti-PF4/P-ABS), over one of the most frequent immune- mediated adverse drug reactions, heparin-induced thrombocytopenia (HIT), to life-threatening autoimmune HIT[2].

PF4 binds to polyanions expressed on the cell wall of a wide variety of Gram-negative and Gram-positive bacteria and plays a role in bacterial host defense[3,4]. Binding of PF4 to polyanions results in its conformational change[5] and exposes the binding site(s) for anti-PF4/P-ABS. As PF4 binds to a large variety of bacteria[4], circulating or boosted anti-PF4/P-ABS can readily opsonize even bacteria the organism has not seen before. These antibodies are associated in the general population[6] with the presence of chronic bacterial infections such as periodontal disease[7]. In medicine, anti-PF4/P-ABS gained major attention as they can induce the severe adverse drug effect HIT[1] when misdirected towards complexes of PF4 and polyanionic drugs, mostly heparin[8]. There is a continuum of clinical sequelae resulting from anti-PF4/P-ABS: the majority of individuals remain asymptomatic[9,10], while in HIT, anti-PF4/P-ABS activate platelets and the clotting system, resulting in life-threatening thrombotic complications when patients receive the polyanion heparin[1,11]. Anti-PF4/P-ABS are also induced after major surgery without heparin treatment when mechanical compression devices are used for thrombosis prophylaxis. In this case, continuous tissue compression likely acts as danger signal[12]. The extreme sequela of anti-PF4/P-ABS is autoimmune HIT, in which individuals develop multiple vessel occlusions without any drug exposure[2,13]. Autoimmune HIT can also be triggered by bacterial infection or major surgery. Typically, in these patients, PF4/P-ABS are found by enzyme-linked immunoassay (EIA) at very high titres and activate platelets in the absence of polyanions in functional assays.

The prothrombotic mechanism of anti-PF4/P-ABS depends on FcγRIIA mediated cell activation. When anti-PF4/P-ABS bind to PF4/P complexes, the resulting immune complexes bind to Fc-receptors on platelets (FcγRIIA)[14]. Cross-linking of FcγRIIA results in platelet activation, leading to the release of more PF4 and formation of further immune complexes, which rapidly recruit other platelets into the prothrombotic process[15]. Activation of the clotting cascade results in thrombin generation and increases the risk for new thrombosis. Polymorphisms of FcγRIIA, which determine its binding affinities to different human IgG subclasses[14] and polymorphisms of signal transduction molecules involved in FcγRIIA signalling[16] have been shown to modulate the risk for clinical HIT.

The antigen to which anti-PF4/P-ABS bind is exposed on PF4 when it forms PF4/P complexes[8] with long polyanions, which force PF4 tetramers into close approximation[17,18]. The antigenic site on a PF4 tetramer is a complex surface formed by at least three PF4 monomers at the closed end of the PF4 tetramer as determined by x-ray crystallography of PF4 in complex with the shortest heparin (fondaparinux) and Fab-fragments of the monoclonal antibody KKO[19]. KKO mimics the biological activity of human anti-PF4/P-ABS (ref. 20), causes HIT in an animal model *in vivo*[21,22], and has been used to understand the binding characteristics of an antibody recognizing PF4/P-complexes and activating platelets[20,23]. In this study with polyclonal human antibodies, the monoclonal KKO served as a standard.

In contrast to the detailed characterization of the PF4/P complexes, and the FcγRIIA-dependent pathway[14,15,24,25], little is known about the features of the antigen-binding part (Fab) of anti-PF4/P-ABS, which determine their biological effects[14,16].

All anti-PF4/P-ABS bind to immobilized PF4/P complexes in EIAs, but only some of them activate platelets in functional assays, for example, the heparin-induced platelet activation assay (HIPA)[26] or the serotonin release assay (SRA)[27,28]. Accordingly, we differentiate three groups of anti-PF4/P-antibodies (all are positive in EIA): group-1 ABS do not activate platelets (negative functional assay, HIPA); group-2 ABS are positive in HIPA but only in the presence of heparin; while group-3 ABS activate platelets even in the absence of heparin. Since group-3 ABS are from the sera of patients who had clinical autoimmune HIT[2], we consider this group as 'autoantibodies'. Exploring the characteristics differentiating anti-PF4(/P) autoantibodies (group-3) from polyanion-dependent antibodies (group-2) bears the potential to better understand mechanisms of antibody-mediated autoimmunity.

In this study, we illustrate that binding characteristics of PF4/P-ABS determine their biological activity. A subset of antibodies from patients with autoimmune HIT (group-3 ABS) cluster PF4 and generate a conformational change in PF4, which allows binding of polyanion-dependent group-2 ABS in a similar way polyanions do. As a consequence, the resulting PF4/group-3 complexes recruit the polyanion-dependent group-2 ABS, which are non-pathogenic in the absence of polyanion treatment (for example, heparin), into the autoimmune process.

## Results

**Characteristics of purified anti-PF4/P antibodies.** We isolated by two-step affinity purification anti-PF4/P-ABS from sera of patients with either (i) no symptoms of HIT (group-1, $n = 5$); (ii) clinical HIT during treatment with heparin (group-2, $n = 5$); or (iii) autoimmune HIT (group-3, $n = 5$); and confirmed their purity by SDS–PAGE (Supplementary Fig. 1). The purity of affinity purified anti-PF4/H ABS was > 95% as tested by reincubating them with PF4/H-coated beads and determining the concentration of the non-PF4/H ABS in the supernatant. As a control, IgG from non-HIT patients' sera were used ($n = 2$). The purified ABS showed similar characteristics as the original serum (Supplementary Table 1). When titrating the antibodies, optical density (OD) increased with increasing concentrations for all antibody groups (Fig. 1a), but with different slopes at concentrations $\leq 20 \, \mu g \, ml^{-1}$ (Fig. 1b). Different OD values indicate that these antibodies have different binding affinities: highest for group-3, followed by group-2 and then group-1 ABS. In the HIPA test, group-1 ABS did not cause platelet aggregation up to a concentration of $89.7 \, \mu g \, ml^{-1}$ (dark cyan, Fig. 2a); group-2 ABS induced platelet aggregation from concentrations $\geq 43.5 \, \mu g \, ml^{-1}$ (blue, Fig. 2a), but only in the presence of heparin (that is, the low-molecular-weight heparin- reviparin); while group-3 ABS induced platelet aggregation from concentrations $\geq 5.2 \, \mu g \, ml^{-1}$ (red, Fig. 2a), independently of heparin.

Scanning electron microscopy (SEM) shows platelet aggregates induced by different antibody groups (at $45 \, \mu g \, ml^{-1}$; Fig. 2a). The small aggregates in the presence of control IgG (Fig. 2b–d) reflect background platelet activation, similar to group-1 ABS (Fig. 2e–g). Group-2 ABS (Fig. 2h–j) caused large (Fig. 2h), less dense aggregates (Fig. 2n; Supplementary Fig. 2) only in the presence of reviparin with 15 min lag-time, while group-3 ABS induced large (Fig. 2k–m), dense aggregates (Fig. 2o; Supplementary Fig. 2) already after 5 min lag-time even in the absence of heparin. The monoclonal antibody KKO ($50 \, \mu g \, ml^{-1}$) induced aggregation of washed human platelets at 5 min lag-time when platelets were coated with $\geq 10 \, \mu g \, ml^{-1}$ PF4, consistent with previous studies[29,30].

**Binding strength of anti-PF4/P antibodies.** We determined the binding strength of the three antibody groups to PF4/H

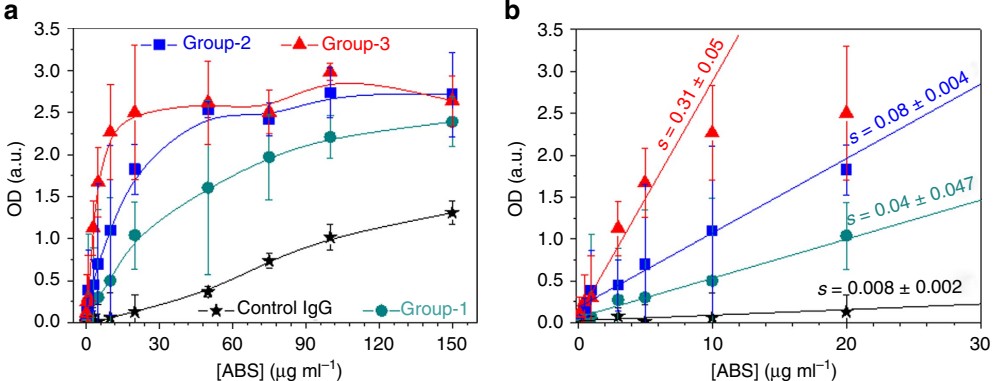

**Figure 1 | Dose-dependent binding of anti-PF4/P-ABS to PF4/H complexes in EIA.** PF4/H complexes were coated to a microtiter plate and binding of ABS to PF4/H complexes was measured by EIA. Mean OD values ± s.d. were averaged from 5 sera for each group and 2 sera for control IgG. (**a**) The curve of control IgG (black stars) provides the background reaction. The specific reactions were lowest for group-1 (dark cyan circles), higher for group-2 (blue squares) and highest for group-3 (red triangles) ABS. (**b**) Slopes (*s*) of the curves obtained in (**a**) for concentrations up to 20 µg ml$^{-1}$ (at which the background for control IgG was an OD up to 0.32).

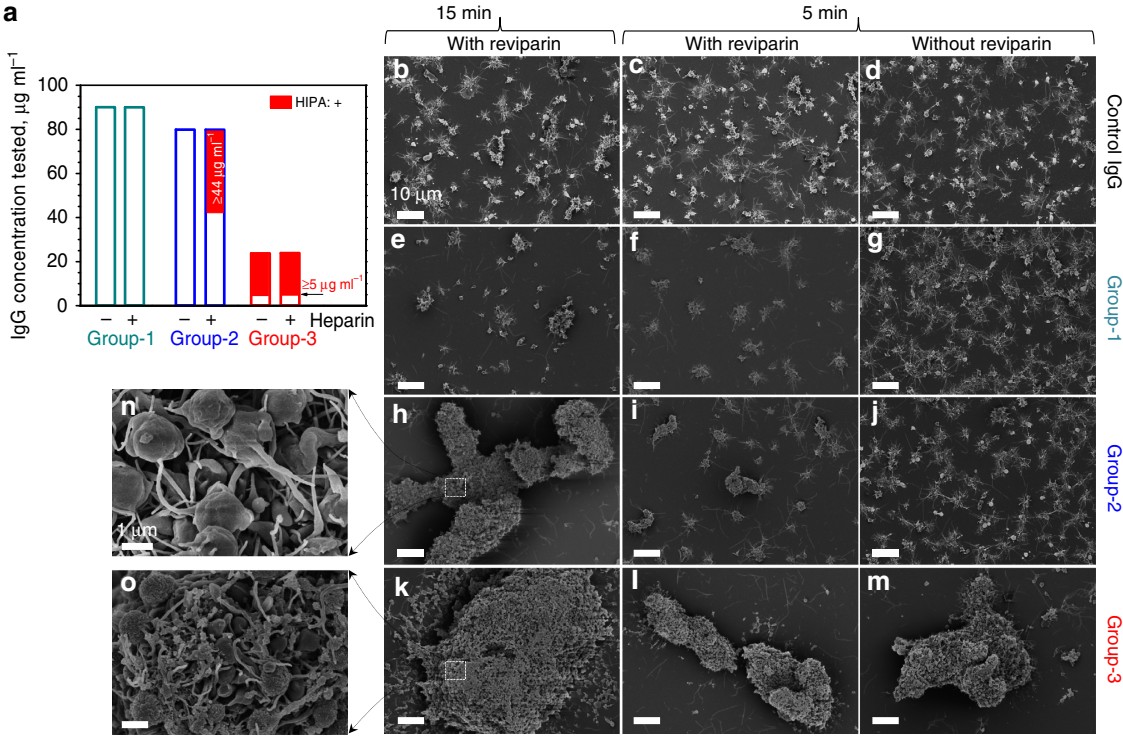

**Figure 2 | Platelet aggregates induced by purified PF4/P-ABS in the HIPA test.** (**a**) Dependence of platelet aggregation on ABS concentration: group-1 ABS (*n* = 5) did not activate platelets, neither in the absence ( − ), nor in the presence ( + ) of reviparin up to a concentration of 89.7 µg ml$^{-1}$ (dark cyan); group-2 ABS (blue; *n* = 5) induced platelet activation (red part) at concentrations ≥ 44 µg ml$^{-1}$ but only in the presence of reviparin. Group-3 ABS (*n* = 5) (red) activated platelets at much lower concentrations (≥ 5 µg ml$^{-1}$) either in the presence or absence of reviparin. (**b–o**) SEM images show detailed platelet aggregates in the presence of different ABS. Only small aggregates occurred in the presence of control IgG (**b–d**) or group-1 ABS (**e–g**), regardless, whether or not reviparin was added. Group-2 ABS induced large platelet aggregates in the presence of reviparin after a lag-time of 15 min (**h**), but not earlier (**i,j**); while group-3 ABS activated platelets within 5 min independently of reviparin (**k–m**). Enlargements show looser platelet aggregates in the presence of group-2 ABS (**n**), but tighter and denser aggregates for group-3 ABS (**o**). Scale bar, 10 µm (**b–m**); 1 µm (**n,o**).

complexes by single molecule-force spectroscopy (SMFS; Fig. 3a–d). We coated one IgG molecule each to different cantilevers. The IgG molecules were obtained from the affinity purified anti-PF4/P IgG fractions of five different sera per group (group-1: *n* = 43 cantilevers; group-2: *n* = 55 cantilevers; and group-3: *n* = 53 cantilevers) and measured their interaction with PF4/H complexes on the substrate. With each cantilever, we recorded 1,000 force–distance (*F–D*) curves and collected the

final rupture force (*F*) for analysis (Fig. 3d). Figure 3e illustrates a representative individual experiment for each antibody group, showing features of rupture force distributions obtained from the specific interactions between antibodies and PF4/H complexes in each set of measurements.

We determined the experimental background as ∼4% across all forces based on the average binding counts of five independent experiments using human control IgG (Fig. 3e, upper panel).

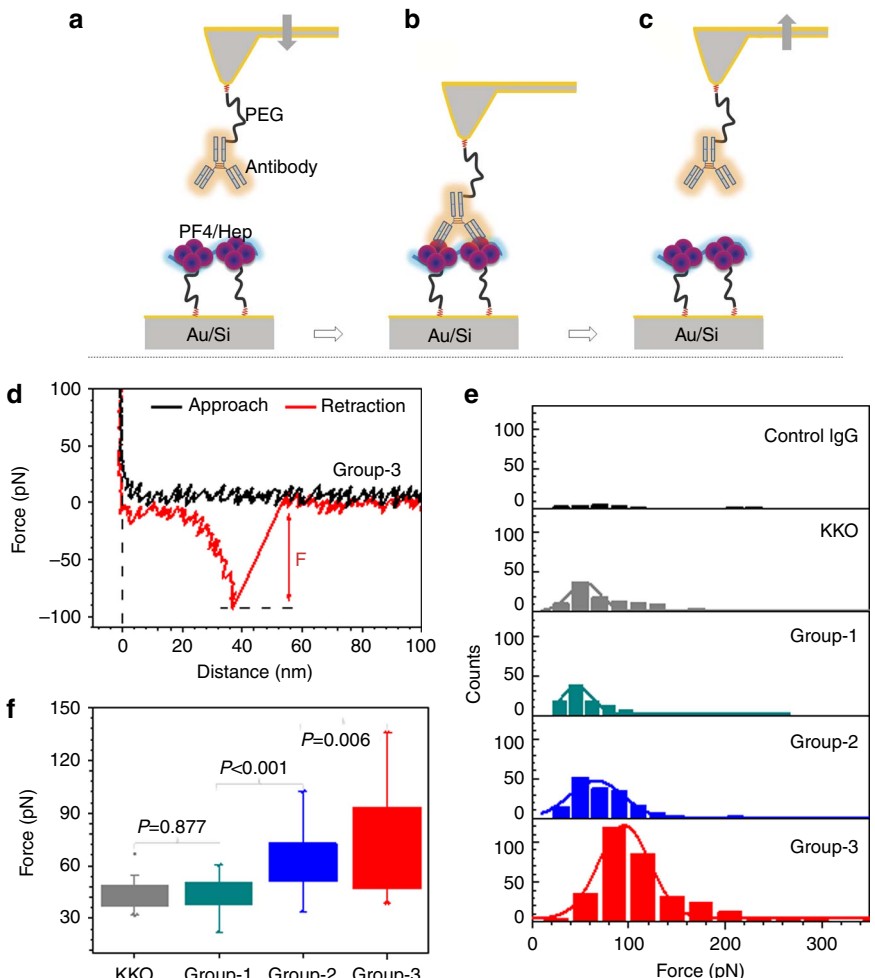

**Figure 3 | Binding strength of antibodies to PF4/H complexes measured by SMFS.** Experimental setup: (**a**) via PEG linkers, a single antibody is attached covalently on an AFM-tip, while the PF4/H complex is immobilized on the substrate. (**b**) The antibody interacts with the PF4/H complex when the tip approaches the substrate. (**c**) When the tip moves away from the substrate, the rupture force of the antibody from the complex is recorded. (**d**) Typical force–distance curve showing the rupture force ($F$) between group-3 ABS and the PF4/H complexes. (**e**) The average rupture force and the corresponding s.e. were determined by a Gaussian fit (solid curves) shown for one representative experiment for human control IgG (black), KKO (grey), group-1 ABS (dark cyan), group-2 ABS (blue) and group-3 ABS (red). (**f**) Average rupture forces and s.d. obtained from different cantilevers for KKO (grey; $n = 18$), group-1 ABS (dark cyan; $n = 43$), group-2 ABS (blue; $n = 54$) and group-3 ABS (red; $n = 51$). Group-2 and group-3 showed higher rupture forces than group-1 ABS and KKO. Rupture force between KKO and group-1 ABS did not differ significantly ($P = 0.877$), while group-2 differ significantly from group-1 ($P < 0.001$) and group-3 differ significantly from group-2 ABS ($P = 0.006$).

For all antibody groups, this 4% were subtracted from their histogram distributions of rupture forces. The probability of specific rupture force events was $\sim 12\%$ for KKO, $\sim 10\%$ for group-1, $\sim 20\%$ for group-2, and $\sim 38\%$ for group-3 ABS. The mean rupture forces and the corresponding s.e.'s were determined by Gaussian fits. To compare binding strengths among antibody groups, the mean rupture forces of each individual experiment from all cantilevers were collected (Fig. 3f). Weakest binding forces were measured for KKO ($43.6 \pm 8.8$ pN, grey) and group-1 ABS ($44.0 \pm 8.1$ pN, dark cyan), they were higher for group-2 ABS ($60.6 \pm 15.4$ pN, blue) and highest for group-3 ABS ($72.4 \pm 26.2$ pN, red). Statistics by analysis of variance (ANOVA) tests showed no significant difference between KKO and group-1 ABS ($P = 0.877$), significant difference between group-1 and group-2 ABS ($P < 0.001$), or between group-2 and group-3 ABS ($P = 0.006$; Fig. 3f). As a blocking experiment, addition of 100 IU heparin, which disrupt PF4/H complexes and partially reverse the conformational change in PF4 induced by low concentrations of heparin[5], resulted in a major reduction of interaction events for all three

antibody groups (Supplementary Fig. 3a,b). Blocking the binding site of PF4/group-3 ABS by group-2 ABS also decreased rupture forces (Supplementary Fig. 3c).

Consistently with SMFS, isothermal titration calorimetry (ITC) results (Supplementary Fig. 4a,c) indicate that group-3 ABS bound to PF4/H complexes with much higher binding energy ($\Delta H = -2.87 \pm 2.06 \times 10^8$ cal mol$^{-1}$) than group-2 ABS ($\Delta H = -2.90 \pm 0.4 \times 10^4$ cal mol$^{-1}$; Supplementary Fig. 4b,c), and their dissociation constant ($K_D$; $\sim 5.3$ nM) was about two orders lower than that of group-2 ABS ($\sim 1.7 \times 10^2$ nM). When PF4 tetramers bind heparin, stable complexes are formed as indicated by high interaction forces of $\sim 150$ pN (ref. 18). This binding force is higher than the strongest binding force measured for group-3 ABS, indicating that the interactions of individual antibodies with PF4/H complexes are not strong enough to disrupt PF4 from PF4/H complexes. Thus, we considered the immobilized PF4/H complexes as one antigen and applied the Bell–Evans model[31,32] to the rupture forces recorded at different loading rates[33]. We found that group-1 ABS have a slightly lower binding affinity ($k_{off} = 15.6$ s$^{-1}$) than group-2 ABS

($k_{off.} = 2.0\,s^{-1}$), or KKO ($k_{off.} = 2.2\,s^{-1}$), while group-3 ABS had the highest binding affinity ($k_{off.} = 0.12\,s^{-1}$; Supplementary Fig. 5). These results indicate that complexes formed by PF4/H complexes with group-3 ABS are more stable than those with group-1 and group-2 ABS, or with KKO.

Since the error bars obtained from the mean values of all individual cantilevers are largely different among antibody groups (Fig. 3f), we plotted the mean force values and their corresponding s.e.'s for each cantilever for more detailed comparison (Fig. 4a–d). KKO (Fig. 4a) and group-1 ABS (Fig. 4b) interacted with rather uniform binding forces with PF4/H complexes as indicated by the relatively small differences among the rupture forces recorded for different cantilevers (most < 60 pN, black dotted line). For group-2 ABS, 40% of all binding forces exceeded 60 pN (22/55 cantilevers; Fig. 4c). For group-3 ABS 44% of all binding forces exceeded 60 pN (23/52 cantilevers) (Fig. 4d), and 15% (8/52) even 100 pN (red dotted line). The low variability in binding forces obtained for the monoclonal antibody KKO and group-1 ABS indicates that they contain homogeneous antibodies (which was expected for the monoclonal antibody KKO), while the heterogeneous binding forces obtained for group-2 and group-3 ABS indicate that the patients sera contained polyclonal mixtures of differently reactive anti-PF4/P-ABS. Group-2 ABS contain also antibodies reacting like group-1 ABS, and group-3 ABS contain also antibodies reacting like group-1 and group-2 ABS beside some super strong reactive antibodies.

The above SMFS experimental design has the drawback that only a limited number of antibodies immobilized on the cantilevers can be tested per serum. To examine whether we immobilized a representative spectrum of these antibodies on the different cantilevers, we performed the reverse experiment (Fig. 4e), in which a PF4/H complex was immobilized on the cantilever, and then brought into contact with the entire fraction of purified PF4/P antibodies immobilized on the substrate. For each purified antibody group, a new substrate was used. This design allowed us to measure the binding forces of PF4/H complexes with the polyclonal PF4/P-ABS mix purified from each single serum. The background was determined using the force histogram for control IgG (black, Fig. 4e) as described above.

KKO (grey) and group-1 ABS (dark cyan) showed histograms with single rupture force distributions, and their average force values are comparable with those in the experiments where the ABS were fixed to the cantilevers shown in Fig. 3e,f. This again specifies that KKO and group-1 ABS contain homogeneously reacting ABS. For group-2 (blue) and group-3 (red) ABS, also the second distribution at high force regimes (arrows, Fig. 4e) was obtained. The first and the second peak correspond to the weak and the strong reactive ABS, respectively (Fig. 4e), indicating again the presence of antibodies with different binding strengths and the strongest binding occurring with group-3 ABS. The mean rupture forces of the second peak were ∼70 pN for group-2 ABS and ∼100 pN for group-3 ABS. As a blocking experiment, the cantilever tip with attached PF4/H complexes was again incubated with the antibodies but now together with high concentrations of heparin (100 IU), which resulted in a strong reduction of the specific rupture events (Supplementary Fig. 3b). Thus, both reverse and the original experiments consistently show that these sera likely contain polyclonal PF4/P-ABS with different binding affinities to PF4/H complexes, and indicate that the capacity of anti-PF4/P-ABS to activate platelets depends on their binding strength to PF4/H complexes.

**Anti-PF4/P-group-3 antibodies cluster PF4.** Next, we asked how group-3 ABS activate platelets in the absence of polyanions. Our hypothesis was that group-3 ABS may self-cluster PF4 leading to PF4/group-3 ABS complexes. This hypothesis was proved by various methodologies:

First, we isolated the antibodies from the different sera using a PF4-column (instead of the PF4/H column). Hardly any antibodies were obtained from control and group-1 sera; group-2 sera showed a minimally increased IgG yield, while high concentrations of antibodies were obtained from group-3 sera. When these antibodies were concentrated to 50 µg ml$^{-1}$, only antibodies purified from group-3 sera activated platelets in the HIPA. The results prove that group-3 sera contain antibodies with PF4 specificity, which activate platelets (Fig. 5h).

Next, we compared the interaction among the antibody groups with PF4 alone by ITC. When the antibodies were tested at the

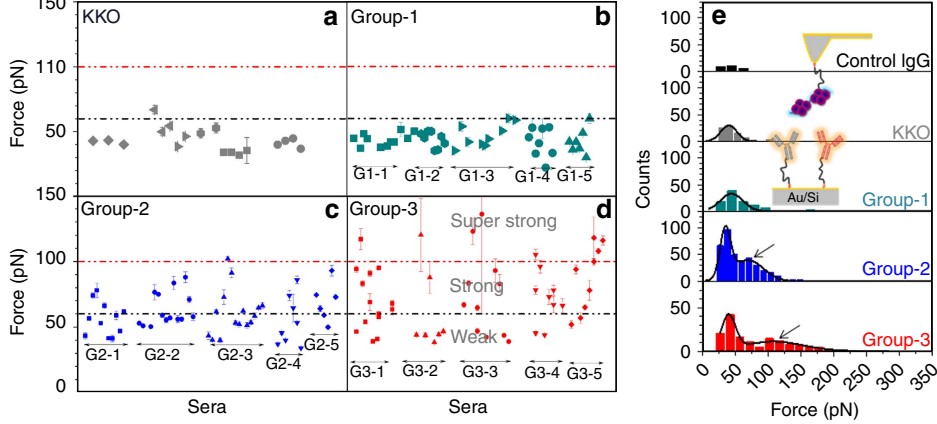

**Figure 4 | Differences in binding characteristics of single antibodies.** To each cantilever, a different antibody purified from the same serum was attached (five sera per group). Each dot shows the mean and s.e. of the rupture forces for each respective antibody. (**a**) KKO and (**b**) group-1 ABS bind to PF4/H complexes with a binding strength mostly lower than 60 pN (black dotted line), while (**c**) group-2 and (**d**) group-3 ABS consist of ABS with different binding forces. A subset of group-3 ABS binds to PF4/H complexes with rupture forces higher than 100 pN (red dotted line). (**e**) Reverse experimental design: a PF4/H complex was immobilized on the tip, while antibodies were immobilized on the substrate (inset). The force histogram for control IgG (black) was used to determine the background. KKO (grey) and group-1 ABS (dark cyan) showed only one force distribution, while two distributions were observed for group-2 (blue) and group-3 (red) ABS. The second force distribution of group-3 ABS shifted to a higher force regime as compared to group-2 ABS (arrows).

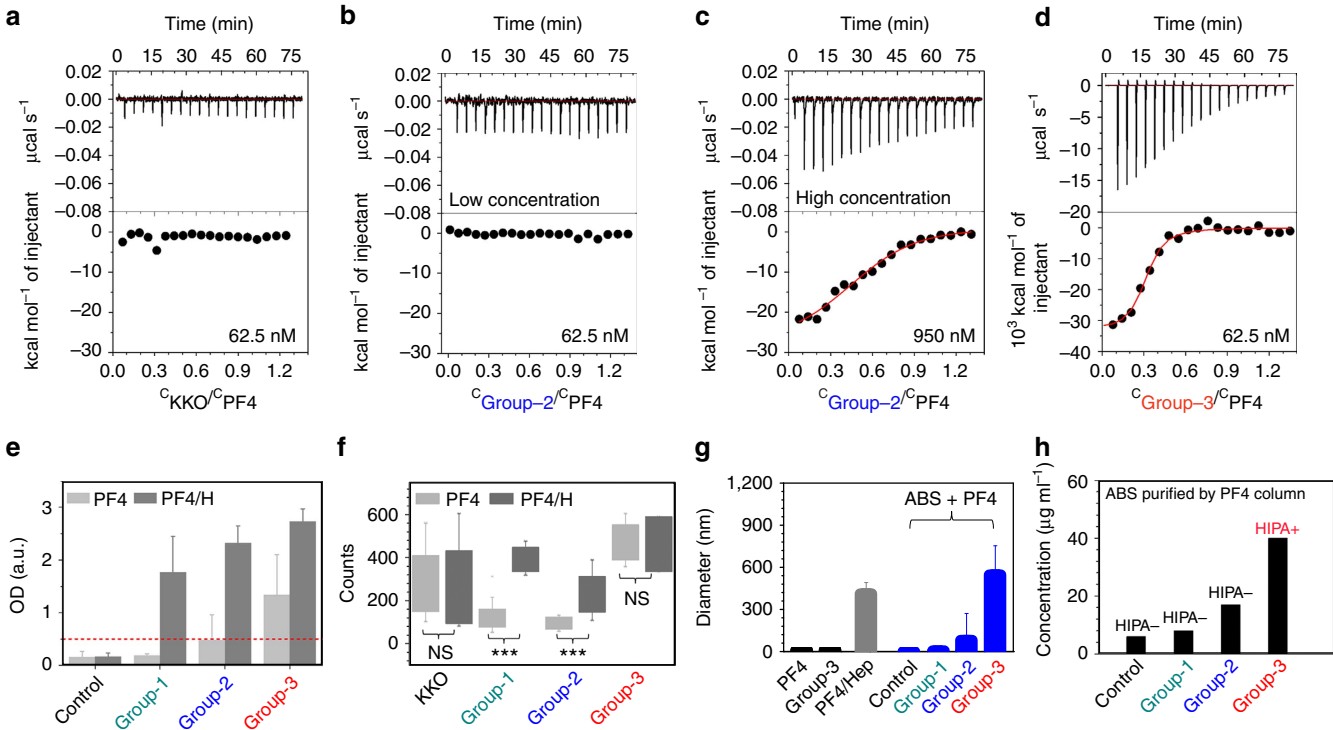

**Figure 5 | Group-3 ABS cluster PF4. (a–d)** Representative binding isotherms for the titration of different antibody groups into the cells containing PF4: raw titration data (upper panel), and integrated heats (lower panel). **(a)** At concentration of 62.5 nM, the thermogram did not change for KKO, or **(b)** group-2 ABS ($n = 2$), but **(d)** clearly changed for group-3 ABS ($n = 2$). **(c)** At higher concentration, slight changes occurred in the thermogram measuring group-2 ABS at 950 nM ($n = 2$), but no changes occurred in the thermogram for KKO at concentrations up to 6.67 μM. The scale for group-3 ABS in **(d)** is 1,000 times larger compared to the scale in **a–c**. **(e)** EIA ODs were lowest for group-1 ABS, higher for group-2 ABS, and highest for group-3 ABS when they interact either with PF4 (light grey) or with PF4/H complexes (dark grey). Binding of all antibodies is strongly enhanced by heparin. The red dotted line indicates the usual OD cutoff of the PF4/H EIA. **(f)** The number of binding counts was largely different when group-1 and group-2 ABS interacted either with PF4 (light grey) or with PF4/H complexes (dark grey) but did not differ for KKO and group-3 ABS. **(g)** DLS results show that PF4 and ABS sizes are smaller than 10 nm (black); when forming complexes with PF4 (blue) group-1 and group-2 ABS did not induce large aggregates while group-3 ABS formed complexes even larger than aggregates formed by heparin (grey). **(h)** The yield of antibodies purified by a PF4 column was highest for group-3 ABS as compared to other antibody groups, and only ABS purified from group-3 sera by the PF4 column were positive in the HIPA. The effluent of the column from group-2 and group-3 sera contained antibodies which activated platelets in the presence of heparin (data not shown).

same concentration of 62.5 nM, KKO (Fig. 5a) and group-2 ABS ($n = 2$; Fig. 5b) did not bind to PF4 alone, while group-3 ABS (purified by PF4/H column) interacted strongly with PF4 (Fig. 5d). The interaction between group-3 ABS and PF4 alone showed two binding sites (stoichiometry $n = C_{ABS}/C_{PF4} = 0.53 \pm 0.003$) indicating that group-3 ABS cluster two PF4 molecules (Supplementary Fig. 4c). When the antibody concentration was increased, KKO still did not bind to PF4 up to a concentration of 6.67 μM, while group-2 ABS showed weak interactions at a concentration of 950 nM (Fig. 5c), which, however, released only 0.1% of the heat compared to group-3 ABS binding.

The above results are fully consistent with the results obtained by EIA (Fig. 5e) with either PF4 (light grey) or PF4/H (dark grey) coated plates: group-1 ABS did not bind to PF4 alone, group-2 ABS showed minimal binding, while group-3 ABS bound to PF4. Still binding of group-3 ABS was higher to PF4/H complexes as compared with PF4 in the EIA. Most likely, all ABS purified from group-3 sera, including polyanion-dependent antibodies, bound in PF4/H EIA, while only the subset of polyanion-independent antibodies in group-3 sera bound in the PF4 EIA (Fig. 5e). By SMFS, group-1 and group-2 ABS showed much less binding events to PF4 than to PF4/H complexes, while the super-reactive ABS belonging to group-3 showed similar binding interactions (Fig. 5f). In addition, the interaction forces of group-3 ABS

purified *via* a PF4-column showed the highest range of binding forces ($\sim 100$ pN) with PF4/H complexes (Supplementary Fig. 6A). These results again indicate that group-3 ABS bind strongly to PF4 alone, while bindings of group-1 and group-2 ABS are heparin-dependent.

Finally, we compared the sizes of complexes formed between different antibody groups and PF4 by dynamic light scattering (DLS; Fig. 5g). PF4/H complexes were used as positive control. Group-3 ABS formed the largest complexes with PF4 as compared to other antibody groups (Fig. 5g) with even larger size than PF4/H.

**Heparin-dependent ABS bind to PF4/group-3 ABS complexes.** The binding energy generated by the interaction of group-3 ABS with PF4 in the ITC experiments ($\Delta H = -3.5 \pm 0.86 \times 10^7$ cal mol$^{-1}$; Supplementary Fig. 4c) was much higher than the energy released when a 16-mer heparin interacts with PF4 ($\Delta H = -7.26 \pm 1.36 \times 10^3$ cal mol$^{-1}$)[5]. As 16-mer heparin can force two PF4 molecules together, based on their high energy release, group-3 ABS most probably also can force two PF4 tetramers together. In addition, the negative entropy of the reaction ($\Delta S = -11.7 \pm 2.8 \times 10^4$ cal mol$^{-1}$ K) suggests that PF4 may change its conformation in the presence of group-3 ABS (Supplementary Fig. 4c). This raised the intriguing hypothesis

that group-3 ABS may cluster two PF4 molecules and hereby, induce a conformational change of PF4 which allows binding of otherwise heparin-dependent ABS.

To prove this hypothesis, we selected cantilevers coated with group-3 ABS showing rupture forces with PF4/H complexes $\geq 80$ pN and incubated them with PF4 in the fluid phase to form PF4/group-3 ABS complexes. The PF4/group-3 ABS complex on the cantilever was then brought into contact for interaction with KKO, group-1, or group-2 ABS coated on the substrates (Fig. 6a/panel-1). The resulting rupture forces were then compared to those obtained when these antibodies coated on the solid substrates interact with PF4/H complexes immobilized on the cantilever (Fig. 6a/panel-2). As expected, we found higher rupture forces for group-2 ABS than for KKO and group-1 ABS (Fig. 6b). Most importantly, the rupture forces from these two experiments were very similar regardless whether the antibodies interacted with PF4/group-3 ABS complexes or with PF4/H complexes. In addition, a bimodal function (two distributions of rupture forces) was obtained when the complexes of PF4/H (top graph, Fig. 6c) or PF4/group-3 ABS (bottom graph, Fig. 6c)

interacted with group-2 ABS immobilized on the substrates, that is, ~70% events $\leq 60$ pN, and ~30% events $\geq 60$ pN consistent with the presence of antibodies with different binding strength. Figure 6d shows the summary of the binding events when the antibodies interact with either PF4, PF4/H or PF4/group-3 ABS complexes at $\geq 60$ pN: KKO, group-1 and group-2 ABS bound weakly to PF4 alone, stronger to PF4/H complexes and also much stronger to PF4/group-3 complexes. These results again underscore that group-3 ABS can cluster PF4 and allow binding of heparin-dependent ABS to PF4/group-3 clusters. Interestingly but not surprising, there seem to be differences between different antibodies: KKO bound stronger to PF4/H complexes than to PF4/group-3 ABS complexes, while group-2 ABS showed the reverse pattern. When binding sites on PF4/group-3 ABS complexes were blocked with group-2 ABS, group-1 and group-2 ABS interacted weaker as shown by the considerable reduction of the specific rupture events (Supplementary Fig. 3c).

To further prove that the PF4/group-3 complexes allow binding of the heparin-dependent group-2 ABS, we performed DLS experiments. The PF4/group-3 complexes were prepared

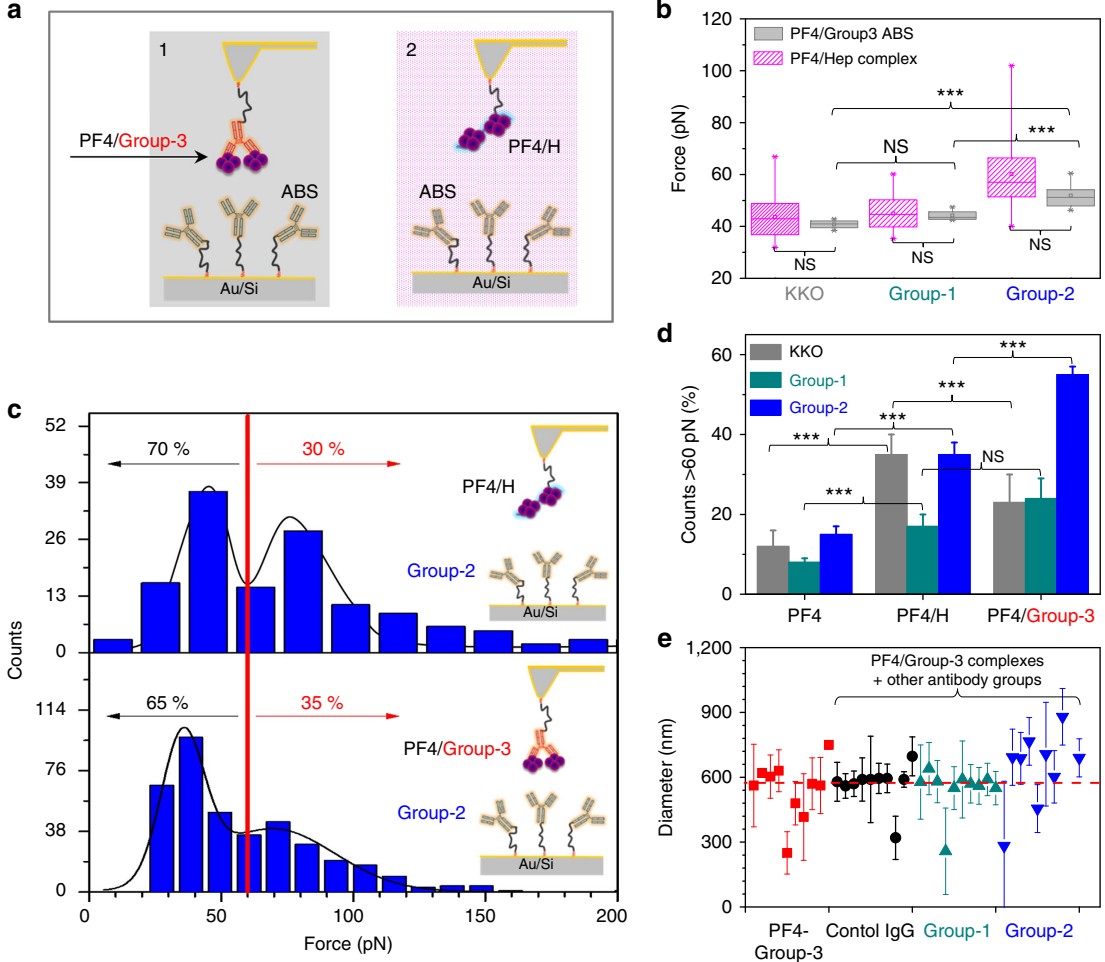

Figure 6 | PF4/group-3 ABS complexes expose the binding epitope for heparin-dependent ABS. (a) Two schematics of either PF4/group-3 ABS complexes (1) or PF4/heparin complexes (2) bound to the tips interacting with KKO, group-1, or group-2 ABS immobilized on the substrates. (b) The rupture forces did not differ between these two systems, indicating that the group-3 ABS could mimic the effect of heparin on PF4 conformation. (c) Force histograms of the interaction of group-2 ABS with PF4/H complexes (top panel); or PF4/group-3 ABS complexes (bottom panel): ~70% of binding forces were $\leq 60$ pN and ~30% of binding forces $\geq 60$ pN ($n = 5$ for each ABS group). (d) The summary counts of interactions with binding forces $> 60$ pN between either PF4, PF4/H, or PF4/group-3 ABS complexes with KKO, group-1, or group-2 ABS again underscores that group-3 ABS can form complexes with PF4 which allow binding of heparin-dependent ABS. (e) DLS results show that when PF4/group-3 complexes (red) were incubated with different antibody groups, no significant changes in the sizes occurred when control IgG (black) or group-1 ABS (dark cyan) were added, complexes increased in size in the presence of group-2 ABS (blue) ($P = 0.047$, ANOVA tested).

before adding other antibody groups (Fig. 6e). When control IgG or group-1 ABS were added to PF4/group-3 complexes, the size of the complexes did not increase significantly. However, by adding group-2 ABS to PF4/group-3ABS complexes, the size increased considerably ($P = 0.047$, one-way ANOVA test) (Fig. 6e).

## Discussion

We show that the biological activity of anti-PF4/P-ABS closely correlates with their binding characteristics. Most antibodies with binding forces to PF4/H complexes $< 60$ pN do not activate platelets, even in the presence of polyanions; antibodies with binding forces between 60 and $\sim 100$ pN activate platelets in the presence of polyanions; while antibodies with binding forces $> 100$ pN activate platelets even in the absence of polyanions.

Remarkably, in contrast to group-1 and group-2 ABS, group-3 ABS bind not only to PF4/H complexes but also to PF4 alone: they can be purified by a PF4 column, bind to PF4 as measured by EIA, SMFS, ITC and DLS, and activate washed platelets in the absence of heparin. This is likely the explanation why group-3 ABS activate platelets in the absence of polyanions.To be able to activate platelets, group-3 ABS and PF4 have to be in close proximity to the platelet surface. It has been proposed that chondroitin sulfate is the binding partner for PF4 on the platelet surface[29]. Group-3 ABS may bind on the platelet surface to native PF4 as well as to PF4, which underwent a conformational change after binding to chondroitin sulfate or polyphosphates[34,35].

Anti-PF4/P-ABS bind with different strengths to PF4/H complexes. Different binding strengths of antibodies to their antigen can be either due to differences in their antigen recognition site or due to binding to their antigen with only one or with both Fab arms. It is well-known that bivalent binding of IgG antibodies largely increases their binding avidity[36], and it has been shown that this is relevant for the pathogenicity of other prothrombotic autoantibodies, for example, against beta2-Glycoprotein 1 in patients with the antiphospholipid syndrome[37]. It is possible that group-1 ABS bind only with one Fab arm to PF4/H complexes, while group-2 and group-3 ABS bind with both Fab arms. However, as there are substantial differences in the binding forces between group-2 and a subset of group-3 ABS, there must be also differences in their binding avidity beyond monovalent or bivalent binding characteristics.

Antibodies obtained from human sera are typically polyclonal, reflected by the different binding characteristics of group-2 and group-3 ABS purified from one serum (Fig. 4). To control for inter-experimental variability in the SMFS experiments, we used the monoclonal antibody KKO as a standard. The relatively low binding forces of KKO to PF4/H complexes in our study were surprising. Binding of KKO to PF4/H complexes depends on the conformation of PF4 (ref. 19). Potentially, under the experimental conditions we used, PF4/H complexes had a slightly different conformation resulting in suboptimal binding of KKO, as compared to the experimental setup of other groups[8,19,20,38]. However, in our hands KKO also strongly activated platelets in the HIPA test. This indicates that besides the binding force, additional characteristics of an antibody are important for its capacity to activate platelets. Interestingly, the thermal off-rates ($k_{off}$) are similar for KKO ($k_{off} = 2.2$ s$^{-1}$) and group-2 ABS ($k_{off} = 2.0$ s$^{-1}$), while it is higher for group-1 ABS ($k_{off} = 15.6$ s$^{-1}$). The low off-rate likely allows long enough binding of the antibodies to platelets to induce platelet activation by cross-linking FcγRIIA. In line with our finding of relatively low binding forces of KKO to PF4/H complexes, previous studies[20,39] had to use KKO in very high concentrations ($\geq 50$ μg ml$^{-1}$ in vitro; 10–20 mg kg$^{-1}$ body weight in vivo) to show any effect. We conclude that the thermal off-rates of the

anti-PF4/P antibodies, which finally determine their affinity, are relevant for their platelet activating capacity.

Sachais et al.[20] reported that KKO is able to cluster PF4 tetramers using precipitation of complexes of radiolabeled PF4 and KKO. Although we did not observe binding of KKO to PF4 alone in the fluid phase, their findings led to our hypothesis[40] that group-3 ABS can cluster PF4 similar to heparin, which subsequently allows group-2 ABS to bind to PF4/group-3 complexes.

Previously, we have shown that polyanions like heparin force two PF4 tetramers together[18], hereby inducing a conformational change in PF4 (refs 5,40), which is consistent with X-ray structure analysis of PF4/fondaparinux complexes[19]. The conformational change of PF4, which is required for binding of group-2 ABS, results in the release of energy[5]. The thermal energy of the interaction of a 16-mer heparin (which is inducing the conformational change allowing group-2 ABS binding to PF4) with PF4 is even lower than the thermal energy of the binding reaction between group-3ABS and PF4 ($\Delta H = -7.26 \pm 1.36 \times 10^3$ cal mol$^{-1}$ (ref. 5) versus $\Delta H = -3.5 \pm 0.86 \times 10^7$ cal mol$^{-1}$) indicating a conformational change of PF4 when group-3 ABS bind. As we could not measure the structure of PF4 in the presence of group-3 ABS, we used the standard immunohaematology approach to prove that PF4 changes its conformation upon complex formation with these ABS, by showing that group-2 ABS bind to PF4/group-3 ABS complexes. First, a single PF4/group-3 ABS complex was immobilized on the cantilever. By SMFS measurements, this complex bound to group-2 ABS with similar binding forces as to PF4/H complexes (Fig. 6a–d). Second, we showed by DLS that the PF4/group-3 ABS complexes grow in size when group-2 ABS are added (Fig. 6e). Third, we adsorbed group-1, and group-2 ABS by PF4/group-3 ABS complexes (Supplementary Fig. 6b). We conclude that group-3 ABS bind to PF4 alone, and thereby, change the structure of PF4 in such a way that group-2 ABS can bind.

In contrast to group-2, group-1 ABS bound to PF4/H and PF4/group-3 complexes but did not increase the size of PF4/group-3 ABS complexes (Fig. 6e). Most probably, the binding strength of group-1 ABS is too weak to further cluster PF4 complexes.

Our experimental approach had several technical challenges. The monoclonal antibody KKO showed some variability of rupture forces in the SMFS experiments. They are most likely explained by a composite of inter-experimental variability and the variability in exposure of the binding epitope for KKO. It is possible that binding of PF4/P complexes to the substrate via a PEG-linker did alter the KKO-binding site[19] on some PF4 molecules. This is further corroborated by the different binding characteristics for KKO, which we observed depending on whether PF4 was presented in the fluid phase (ITC experiments) or on the solid phase (SMFS experiments). In addition, Litvinov et al.[23] found slightly higher binding forces of KKO than we did. However, we used a pull-off speed of 1,000 nm s$^{-1}$, while they used 1,600 nm s$^{-1}$ (ref. 23), which results in higher rupture forces[36,41–44]. In addition, higher PF4 concentrations (1 mg ml$^{-1}$) also result in an increase in the binding strength of KKO to PF4 (ref. 23). To avoid multiple rupture artifacts, we did not use such a high concentration.

The PF4 tetramer consists of four monomers and the complexes of heparin and at least two PF4 tetramers are highly complex structures. This raises the issue, whether the rupture forces measured are of bonds between antibodies and the antigen or bonds within the complex antigen. Our results and observations strongly indicate that under the experimental conditions used, rupturing of PF4 monomers or PF4 tetramers from PF4/H complexes did not occur. First, we have previously determined a

high melting temperature of PF4 tetramers ($\sim 96\,°C$)[45], which suggests that PF4 tetramers are stable. Second, we have shown before that the probability of rupture forces of PF4-heparin bonds are $\sim 150\,pN$ (ref. 18), which is higher than the probability of rupture forces of the most reactive group-3 ABS (in the range of 80–140 pN). Finally, if PF4 monomers would rupture from the tetramer, they would bind to the antibodies on the tip, blocking the antibody binding sites, and further specific rupture events could not be measured due to highly repulsive forces[18]. However, this was not the case since the frequency of specific rupture events was almost constant from beginning to the end of 1,000 measurements. To exclude the possibility of multiple antibodies bound to the cantilever, we only analysed force–distance (F–D) curves with one rupture step[23]. Binding of more than one antibody exhibits a sequence of rupturing steps, as it is nearly impossible that several antibodies rupture at exactly the same distance due to the pulling geometry[46]. We show that highly reactive autoantibodies against PF4 change this endogenous protein in a way that typically polyanion-dependent ABS can bind. Consequently, PF4/group-3 ABS complexes recruit non-pathogenic anti-PF4/P-ABS, which enhances formation of immune complexes and activation of platelets (in the absence of heparin), finally resulting in the clinical presentation of autoimmune HIT and life-threatening thrombotic complications[2,47]. However, our in vitro studies do not prove whether group-3 ABS show the same behaviour in recruiting group-2 ABS also in vivo. Generation of a monoclonal antibody showing the characteristics of group-3 ABS would finally allow confirming our hypothesis by animal experiments.

Transferring the mechanism of the anti-PF4 autoimmune process to other autoimmune disorders may explain an unresolved issue regarding the pathogenicity of certain auto-antibodies. In nearly all antibody-mediated autoimmune diseases, the antibody specificities found in severely affected patients are also found in asymptomatic individuals of the general population[48,49]. It is an intriguing hypothesis that many of these antibodies are not the pathogenic antibodies themselves. Rather, they can be recruited into the autoimmune process in the presence of small amounts of additional antibodies showing similar features as the group-3 ABS we describe in the present study, that is, they are able to change the structure of an endogenous protein thereby expressing an epitope, which allows recruitment of otherwise non-pathogenic antibodies.

## Methods

**Ethics.** The use of human sera obtained from healthy volunteers and patients with HIT including the informed consent procedure was approved by the ethics board at the University of Greifswald.

**Antibody purification.** Anti-PF4/H antibodies were isolated from five patient sera per group by two-step affinity chromatography, that is, first by a protein G column to isolate total IgG and then by a PF4/H column (or PF4 column) to extract only anti-PF4/P-ABS.

*Protein G column antibody purification.* Two millilitre protein G-coated sepharose beads (GE Healthcare Europe GmbH, Freiburg, Germany) were washed three times with PBS at 4°, 300g for 5 min. Patient sera were three times diluted in protein G IgG binding buffer (BB; Thermo Fisher Scientific, Darmstadt, Germany), and then filtered using a 450 nm membrane (Whatman, Dessen, Germany). The sera were then transferred to a protein G column and incubated at room temperature (RT) for 1 h before extensive rinsing with 30 ml BB. Total IgG was eluted from the column with 0.1 M glycine, pH 2.7 and neutralized by Tris buffer pH 8.0 (Thermo Fisher Scientific, Darmstadt, Germany).

*PF4/Heparin column antibody purification.* Biotinylated PF4 (bPF4) as either purchased (Chromatec, Greifswald, Germany) or self-biotinylated[50]. Briefly, PF4 (Chromatec, Greifswald, Germany) was incubated with heparin sepharose (GE Healthcare, Solingen, Germany) for 30 min at RT and then at 4°C overnight. Then, the PF4/heparin sepharoses were incubated with biotin-XX sulfosuccinimidyl ester (Life Technologies, Darmstadt, Germany) for 1 h at RT. After washing with PBS to remove free biotin, the biotinylated PF4 was eluted with high salt buffer (2 M) and dialysed against PBS before use.

Complexes of 0.5 IU ml$^{-1}$ UFH with 20 µg ml$^{-1}$ mixture of bPF4 and PF4 (30% bPF4:70% PF4) were formed in PBS at RT for 1 h. A suspension of 2 ml streptavidin-coated sepharose beads (GE Healthcare Europe GmbH, Freiburg, Germany) was washed three times with PBS at 4°C, 300g for 5 min. The reaction between bPF4 in the bPF4/PF4 complexes and streptavidin-coated sepharose beads was carried out by mixing the complexes with the beads for 30 min at RT and incubating further at 4° overnight. The sample column was then washed five times with PBS to remove unbound molecules, and then the IgG fraction collected from protein G column was transferred to the PF4/H column, and gently stirred at 4° overnight. After washing, the bound antibodies were eluted with 0.1 M Glycine buffer, pH 2.7 and neutralized by Tris buffer, pH 8.0. Antibody concentrations were determined by IgG absorbance at 280 nm using a Nanodrop 2000c spectrophotometer (Thermo Scientific, Wilmington, USA).

*PF4 column.* Biotinylated human platelet factor 4 (bPF4; Chromatec, Greifswald, Germany) was incubated with total IgG from different antibody groups in PBS at RT for 1 h and then transferred to 2 ml streptavidin-coated sepharose beads for 30 min at RT, and then incubated further at 4° overnight. After rinsing, the bound PF4 antibodies were eluted with 0.1 M Glycine buffer, pH 2.7 and neutralized by Tris buffer, pH 8.0. Antibodies were concentrated to 50 µg ml$^{-1}$ for HIPA tests.

**SDS–PAGE.** To check the purities of ABS, 15 µl of each sample (100 µg ml$^{-1}$) were mixed with 5 µl of 2× Tris-Glycine SDS sample buffer using non-reducing or reducing SDS–PAGE (Invitrogen, Carlsbad, CA). For reducing conditions, the sample was incubated at 94°C to separate heavy and light chains. After that, 15 µl of the sample was loaded into the wells of a 4–20% Tris-Glycine mini gel (Life Technologies GmbH, Darmstadt, Germany). A marker of 170 kDa molecular weight was used as a standard sample. The gel was run at 75 V for stacking of samples and then at 110 V until the dye front reached the bottom of the gels. The running buffer contains 0.025 M Tris, 0.19 M glycine, 0.1% SDS, pH 8.6. The gels were stained with colloidal blue staining kit (Invitrogen, Carlsbad, CA) and then rinsed using Milli-Q water. To check the specificities of the affinity purified anti-PF4/P-ABS, we adsorbed the purified antibodies with PF4/H-coated beads, by which 95.4% of the purified antibodies could be adsorbed.

**PF4 and PF4/heparin enzyme immunosorbent assay.** PF4 or PF4/H complexes (20 µg ml$^{-1}$) were immobilized on a microtiter plate overnight at 4°C (ref. 51). Then, purified antibodies at different concentrations were incubated with PF4 or PF4/H complexes on microtiter plate for 1 h at RT. After washing with buffer (150 mM NaCl, 0.1% Tween 20, pH 7.5), the samples were incubated with peroxidase-conjugated goat anti-human IgG (1: 20,000, Dianova, Hamburg, Germany) for 1 h at RT. That substrate yields a visible colour change which indicates the binding of human antibodies to the coated PF4/heparin complexes. Bound antibodies were subsequently detected by measuring the OD at a wavelength of 450 nm.

**Heparin-induced platelet activation assay (HIPA).** For the HIPA test, 75 µl of washed platelets were incubated with 20 µl sera or purified antibody fractions of different concentrations with either the low-molecular-weight heparin (reviparin) 0.2 IU ml$^{-1}$ or platelet suspension buffer added[26]. High heparin concentration (100 IU ml$^{-1}$) was used to show heparin dependency of the reaction. KKO was used at a concentration of 50 µg ml$^{-1}$ (ref. 39) and 10 µg ml$^{-1}$ PF4 were added.

**SEM imaging of platelets.** Twenty-four millimetre round glass coverslips or silicon chips (Plano GmbH, Wetzlar, Germany) were cleaned using a standard RCA cleaning procedure (a 1:1:5 solution of NH$_4$OH (ammonium hydroxide):H$_2$O$_2$ (hydrogen peroxide):H$_2$O (water) at 70°C for 10 min)[52]. The RCA cleaned glass coverslips were dried under nitrogen stream, cleaned for 10 min with oxygen plasma (600 W, oxygen flow 500 s.c.c.m., Gigabatch 310, PVA TePla, Germany). The freshly cleaned substrates were then incubated for 3 h at 37° in 50 µg ml$^{-1}$ collagen G type I (Biochrom GmbH, Berlin, Germany). Subsequently, the coated substrates were rinsed three times with PBS, and dried under nitrogen stream. Then, aliquots of 10 µl of platelets were taken from the reaction wells of the HIPA test after 5 and 15 min, in which platelets of healthy donors were incubated with the anti-PF4/P-ABS (final concentration 45 µg ml$^{-1}$) purified from patients sera (group-1, group-2, group-3 ABS or the IgG fractions of healthy volunteers as controls). To allow for a comprehensive overview, the samples were dropped on the collagen-coated substrate and kept for 30 min at RT. After washing, platelets were fixed with 2.5% glutaraldehyde (Sigma-Aldrich, Munich, Germany) in PBS for 20 min followed by post-fixation with 1% OsO4 (Sigma-Aldrich, Munich, Germany) in PBS for 20 min. Subsequently, platelets were dehydrated for 5 min/each in ethanol solutions from 30 to 100%. Finally, samples were dried in Polaron Critical Point Dryer (Quorum Technologies Ltd, Kent, UK) before coating with gold in sputter coater for SEM imaging.

**SMFS.** To measure binding strength between antibodies and PF4/H complexes, antibodies were attached covalently to the AFM-tips via a PEG linker, while the PF4/H complexes were immobilized on the substrate using amide bond coupling via a PEG linker. The long PEG linkers were used to increase the flexibility and

mobility of the molecules between tip and substrate and to reduce the chance that more than one antibody or PF4/H complex bind to the tip.

**Immobilization of PF4/heparin complexes.** Gold-coated silicon chips (Plano GmbH, Wetzlar, Germany) were exposed to oxygen plasma for 30 min to remove organic components and to generate a hydrophobic surface before incubating with thiol-PEG-carboxylic acid (HS-PEG-COOH, PEG Mw 3,400 Da, Nanocs, USA) for 2 h at RT. After that, carboxyl groups at the end of PEG linkers were activated by amine coupling kit EDC:NHS (Biacore, Uppsala, Sweden)[18]. PF4/H complexes were preformed by a mixture of 20 µg ml$^{-1}$ PF4 with 0.5 IU ml$^{-1}$ heparin (Heparin-Natrium-25,000, Ratiopharm GmbH, Ulm, Germany) in PBS for 1 h at RT[53]. The complexes were then transferred to the PEG-coated substrate for another 1 h at RT. Finally, free activated groups on the surfaces were blocked by adding 1.0 M ethanolamine (Biacore, Uppsala, Sweden) for 1 h at RT. Samples were finally rinsed with PBS and used within a day. For immobilization of PF4 on the substrate, the same protocol as used for immobilization of PF4/H complexes above was used.

To control functionality of surface chemistry, the PF4/H complexes coated substrate was incubated with anti-PF4/H ABS, and then incubated with Alexa Fluor 488 GaH IgG Atto488 fluorescent antibodies (Biotium, Hayward, CA, USA) for 1 h at RT and finally rinsed with PBS. The integrity of PF4/H complexes was controlled by assessing binding of anti-PF4/H antibodies to the coated PF4/H complexes on the substrates by confocal laser scanning microscopy (Supplementary Fig. 7).

**Functionalization of the AFM tip with antibodies.** Both sides gold-coated silicon nitride cantilevers with a nominal spring constant of 6 pN nm$^{-1}$ (Olympus Bio-lever, Tokyo, Japan) were coated with thiol-PEG-carboxylic acid and the carboxyl groups at the end of PEG linkers were activated as described above[18,54]. Next, the cantilevers were incubated for 1 h at RT with the different antibodies (70 µg ml$^{-1}$): that is, KKO (BIOZOL, München, Germany), control IgG, or purified human anti-PF4/P-ABS (pre-dialyzed against PBS). KKO was used as a standard, and human IgG purified from sera of healthy volunteers, testing negative in both PF4/H EIA and HIPA was used as negative control. When titrating different antibody concentrations for coating the surfaces (or the cantilever tips), multiple rupture events were observed at concentrations > 70 µg ml$^{-1}$, while at much lower concentrations, less specific ruptures events were obtained. Therefore, we used 70 µg ml$^{-1}$ as an optimal concentration for coating tips and substrates. Finally, free activated groups on the surfaces were blocked by adding 1.0 M ethanolamine (Biacore, Uppsala, Sweden) for 1 h at RT. After rinsing with PBS, the cantilevers were immediately used for SMFS measurements; otherwise, they were kept at 4 °C until use (within three days).

To form PF4/group-3 ABS complexes on the cantilever, the group-3 ABS were immobilized as described above. Then, the cantilevers coated with group-3 ABS showing strong reactivity with PF4/H complexes as determined by rupture force measurements were incubated with PF4 (70 µg ml$^{-1}$) in PBS for 1 h at RT. The cantilevers were then rinsed with PBS before measuring their interactions with other antibody groups (KKO, group-1 and group-2) coated on the substrates.

**Measurement of force–distance curves.** The measurements were carried out in PBS using JPK NanoWizard 3 (Berlin, Germany). Before each experiment, the cantilever spring constant was independently measured by a thermal tune procedure[55,56]. Due to the complexity of three involved molecular interactions, that is, among PF4, heparin, and antibody; or among PF4, group-3 ABS, and other antibodies, the antibody-functionalized tip was brought into contact with the PF4/H complexes coated the substrate and allowed to rest for 1 s for interaction[57,58]. To overcome any electrostatic appearance between tip and substrate, a setpoint of 300 pN was used to control the maximal force of the tip against the surface[16]. Under these conditions, control IgG showed 4% rupture events. We, therefore, considered the 4% rupture events as background and always subtracted this background from the force histograms of the PF4/P antibodies. To compare the change in the binding force among antibodies, 1,000 force–distance (F–D) curves were recorded at the same tip velocities (1,000 nm s$^{-1}$) for each experiment. Different velocities (ranging from 10 to 4,000 nm s$^{-1}$) were applied to explore the binding kinetics of the antibodies. For each serum, the measurements were repeated at least three times using independently prepared cantilevers and freshly prepared PF4/H-coated substrates.

For disrupting/blocking binding sites on PF4/H or PF4/group-3 complexes, their interactions with antibodies were first measured. After that, heparin (final 100 IU) or group-2 ABS (final 50 µg ml$^{-1}$) were added to the liquid cell containing PF4/H or PF4/group-3, respectively, and incubated for 30 min before re-measuring. The rupture forces before and after adding heparin or group-2 ABS were compared.

**Data analysis.** The rupture forces at the final rupture points before the cantilevers went back to the rest position were collected from F–D curves exhibiting the behaviour of polymer stretching using the JPK data processing software (version 4.4.18 +). The mean rupture force values and their corresponding errors were determined by applying Gaussian fits to the data using Origin software (version

8.6). The one-way ANOVA was used to determine significant differences between binding strengths of different antibody groups.

To determine thermal off-rates, the cantilevers were coated with homogeneous group-1 ABS or KKO as described above, while the cantilevers coated with group-2 or group-3 ABS were pre-tested to select uniformly strong reactive ABS, that is, with an interaction force of ~70 pN for group-2 ABS and ~90 pN for group-3 ABS. The mean rupture forces were determined by applying Gaussian fits and then the mean rupture force of each measurement measured from three sera per antibody groups were averaged to obtain medians and s.d.'s.

**Anti-PF4/P-ABS binding kinetics.** To understand the kinetics of the bonds formed between the three antibody groups and PF4/H complexes, thermal off-rates (dissociation constant) of the bonds between antibodies and PF4/H complexes were determined by measuring the rupture forces at different loading rates and applied the Bell–Evans model[36,43,44]. The model described that the rupture force increases proportionally to the natural logarithm of the loading rate during retraction. The relation between rupture force and loading rate ($\dot{F}$) is described as:

$$F(\dot{F}) = \frac{k_B T}{\Delta x} \ln\left( \frac{\dot{F}}{k_{off}} \frac{\Delta x}{k_B T} \right) \tag{1}$$

were the involved parameters are: the Boltzmann constant ($k_B$), the experimental temperature ($T$, $k_B T = 4.1$ pN nm), the distance from the bound to the unbound state ($\Delta x = k_B T/m$), the slope of the fit ($m$), the thermal off-rate ($k_{off} = \dot{F}(F=0)\Delta x/k_B T$), the loading rate ($\dot{F} = v k_{eff}$), the pulling speed ($v$) and the effective spring constant of the system composed of the springs of both cantilever and involved linkers ($k_{eff}$). For the Bell–Evans model, the PF4/H complexes were regarded as one molecule, as these complexes are very stable and do not dissociate during the experiment.

**ITC.** Before measurement, PF4 and antibodies were dialysed overnight in PBS, pH 7.4. ITC measurements were carried out using a MicroCal iTC200 calorimeter (Malvern Instruments, Malvern, UK). PF4 in PBS (34.4 to 625.0 nM) was added to the sample cell and a solution of antibodies (62.5 nM to 6.67 µM) was loaded into the injection syringe. The antibody solution in the syringe was added by 19 injections of 2 µl each into PF4 in the cell at 25 °C, 1,000 r.p.m. stirring with 240 s intervals. Control titration was performed by injecting antibodies into PBS buffer and subtracted before the data analysis. The area under each peak of the resulting heat profile was integrated, normalized by the concentrations, and plotted against the molar ratios of antibodies to PF4 using an Origin script supplied with the instrument (Origin 7, Northampton, MA, USA). The one set of sites model was used to fit the binding isotherm data with non-linear regression to determine thermodynamic parameters such as the stoichiometry of the interaction ($n = C_{ABS}/C_{PF4}$, where $C$ is the concentration in mol l$^{-1}$), the equilibrium constant ($K_A$), change in enthalpy ($\Delta H$), and change in entropy ($\Delta S$) of the interactions. The s.d. were obtained by two to three experimental repetitions.

**DLS.** PF4 and antibodies were centrifuged at 4 °C for 20 min at $14.8 \times 10^3$ g min$^{-1}$ to remove aggregates if any. To form complexes of PF4 with antibodies, PF4 (5 µg ml$^{-1}$ final) was incubated with affinity purified antibodies (10 µg ml$^{-1}$ final) at RT for 30 min before measurements (Supplementary Fig. 6c). To investigate whether control IgG, group-1, group-2, or group-3 ABS bind to PF4/group-3 ABS complexes, preformed PF4/group-3 ABS complexes were incubated with these antibodies (10 µg ml$^{-1}$ final) for 30 min at RT before measurements. DLS-fixed scattering angle Zetasizer Nano-S system (Malvern Instruments Ltd., Malvern, UK) and disposal cuvettes were used. The Z-average size of PF4, antibodies or their complexes in terms of the hydrodynamic diameter were measured in PBS at 25 °C and light scattering was detected at 173° using 10 repeating measurements. Data analysis was performed using the Zetasizer software, Version 7.11 (Malvern Instruments Ltd., Malvern, UK).

**Data availability.** The data that support the findings of this study are available from the corresponding authors upon reasonable request.

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

## Acknowledgements

We thank Ulrike Strobel, Ricarda Raschke, Krystin Krauel, Jessica Fuhrmann for providing washed platelets and help with EIA experiments, Doreen Biedenweg for the help with SDS–PAGE and Raghavendra Palankar for the help with immunofluorescence tests. This work was supported by the Deutsche Forschungsgemeinschaft (DFG, Germany; NG 133/1-1), the European Research Council (ERC) Starting Grant 'PredicTOOL' (637877) Starting up Project FOMM/FOCM-2016- 04 University Medicine Greifswald and the German Ministry of Education and Research (BMBF) within the project FKZ 03Z2CN11.

## Author contributions

T.-H.N. designed and performed the experiments, analysed and interpreted the data and wrote the manuscript. A.G. developed the study concept, designed the experiments, provided the patient samples, interpreted the data and wrote the manuscript. N.M. performed the electron microscopy studies. M.D. discussed the data and wrote the manuscript. All authors read and agreed to the final version of the manuscript.

## Additional information

**Competing interests:** The authors declare no competing financial interests.

