## [Peer Review File · Nature Communications]

Reviewers' comments:

Reviewer #1 (expert in HIT)

Remarks to the Author:

Studies were done to define how functional/biologic properties of antibodies produced by patients treated with heparin correlate with clinical manifestations, ranging from no symptoms to severe thrombocytopenia. Patient antibodies (abs) studied were broken down into three groups. All three types of antibody react with heparin/platelet factor 4 complexes. Group 1 abs are negative in standard platelet activation assays. Group 2 abs activate platelets when low dose heparin is present. Group 3 abs activate platelets in the absence of heparin and are thought to cause more severe disease (thrombocytopenia, thrombosis) than the other types, even in patients not given heparin. Findings made are interpreted as showing that behavior of the 3 types of abs can be explained on the basis of their affinity/avidity for heparin/PF4 complexes (Group 3 > Group 2 > Group 1. This is an important finding because, although it is generally thought that Group 2 and 3 antibodies are more likely to cause disease than the much more common Group 1 antibodies, the molecular basis for this difference is not well understood. ON the basis of findings made using SMFS, the authors further suggest that Type 3 abs bind to PF4 and modify the chemokine in such a way that it is recognized by Type 1 and Type 2 abs even in the absence of heparin and that this leads to "recruitment" of Type 2 and Type 1 abs into an "autoimmune process." It is proposed that findings made may have implications for other types of autoimmune disease.

Major comments

1. The authors suggest that Group 3 abs are "autoantibodies" and that they act like heparin in the sense that they aggregate PF4 to produce complexes that are then recognized by Group 2 and perhaps Group 1 abs, thus "recruiting" the latter abs "...into an autoimmune process." The SMFS studies appear to be consistent with this possibility but considerable doubt can be raised as to whether single molecule observations like these can be extrapolated to what may be happening in a patient who has a "Group 3" antibody. There are various ways to track formation of large IgG/PF4 aggregates in solution (e.g., light scattering). Why don't the authors incubate their purified HIT antibodies with PF4 in solution at various concentrations and see whether larger aggregates actually do form in the absence of heparin when Group 3, but not Group 2 or Group 1 abs are present? They could even test directly whether Group 3 abs recruit Group 2 abs into the complexes. Alternatively, they could plate a Group 3 ab, allow it capture PF4, see whether this creates a target recognized by Group 2 and Group 1 abs. Positive findings would greatly strengthen claims made on the basis of the SMFS findings.

2. Line 204: "Thus, both reverse and non-reverse experiments consistently show that the original sera likely contain polyclonal ABS with different binding affinities to PF4/H complexes and indicate that the capacity of anti-PF4/P-ABS to activate platelets depends on their binding strength to PF4/H complexes."

The authors' data convincingly show that Group 3 abs recognize PF4/P complexes with higher affinity than Group 2 or 1 abs and that the former abs activate normal platelets even in the absence of low dose heparin. These findings are important, but It doesn't necessarily follow that the correlations reflect a cause-and-effect relationship. The authors' conclusions would be strengthened if they could identify what is being recognized on the platelet surface by Group 3 abs in the absence of heparin and by the Group 2 abs in the presence of heparin in the HIPA studies (Figure 2). This question is very important. Surely it at least deserves some discussion.

3. Line 292: "Previously, we have shown that polyanions of certain length and charge density induce

the conformational change in PF4 (5, 28), required for binding of anti-PF4/P-ABS, by forcing two PF4 together (16). This results in fusion of the positive charge cloud of two PF4 tetramers and the release of additional energy, required to switch the structure of the PF4 protein for expression of the neo-epitope (16).

References 5 and 28 do show that various polyanionic compounds induce structural changes in PF4 but did not directly demonstrate any connection between these changes and HIT antibody binding. In Ref 16, it was shown that PF4 molecules repel each other because of their strong positive charges but that when PF4 is treated with glutaraldehyde to cross-link PF4 monomers "internally," weak PF4-PF4 interaction can be demonstrated by SMFS. The authors suggest this finding supports the view that PF4-PF4 bonds exist in clusters of PF4 aggregated by "long" heparin molecules. In the view of this reviewer, none of these data support the authors' contention (expressed in lines 292-296 and in many other places in the text) that Group 3 HIT antibodies can induce changes in PF4 that lead to ".....expression of the neoantigen" recognized by Group 1 and Group 2 antibodies and in this sense can "substitute for heparin." The SMFS studies are consistent with this possibility but it is a real stretch to suggest that these findings can be used to draw conclusions about what happens in a patient.

4. The EM studies of the HIPA reactions shown in Figure 2 are impressive. Were these studies done with patient serum or with purified antibodies? Were any HIPA studies done with the purified antibodies to confirm that their reactions mirror those obtained with the original serum?

5. Supplementary Fig 1: It is unusual to be able to elute antigen-specific IgG from serum in quantities sufficient to produce the bands shown in Suppl Fig 1. Were any studies done to determine experimentally what fraction of IgG present in the bands shown in lanes 9-11 is actually antibody specific for PF4/P? The gels could, for example, be performed before and after absorbing the eluates with beads coated with PF4/heparin complexes.

6. The Discussion is long and rambling and contains many seemingly disconnected paragraphs. An example is the last 4 paragraphs running from lines 296-317. The most interesting and important finding described in this paper is the convincing correlation between the functional/biological properties of the different HIT antibody groups and their affinity/avidity for heparin/PF4 complexes. The paper could be greatly improved by focusing on this and a few other significant findings that are well-documented and by avoiding undue speculation about the significance of the SMFS findings for HIT pathogenesis and for autoimmune diseases in general.

Minor comments:

1. Lines 35-36. "HIT occurs when anti-PF4/P-antibodies activate PF4/heparin-coated platelets; while...." This reviewer is unaware of any published evidence showing that HIT is caused by such a process.

2. Lines 55-56. "Binding of PF4 to polyanions results in its conformational change (5, 6) exposing the binding site(s) for anti-PF4/P-ABS. Reference 6 is inappropriate because the only antibody studied was KKO, which binds to PF4 in the absence of heparin, and the only polyanion studied was fondiparinux, which induces little or no conformational change in PF4.

3. Lines 267-269. "We show for the immune response against PF4/P complexes that the biological activity of anti-PF4/P-ABS depends on their binding affinity, as measured by binding forces of highly purified anti-PF4/P-ABS." It would be more accurate to say that biologic activity "correlates closely" with binding affinity since "cause-and effect" was not documented. And the term "immune response"

is inappropriate because the paper does not deal with this subject (induction of an HIT antibody response).

4. Lines 310-311. "In nearly all antibody-mediated autoimmune diseases, the antibody specificities found in severely affected patients are also found in asymptomatic individuals of the general population." Can the authors provide a reference for this generalization?

5. Lines 320-322. "It has been shown for other prothrombotic autoantibodies, e.g. against β 2 Glycoprotein 1 in patients with the antiphospholipid syndrome that bivalent binding of autoantibodies largely increases their binding avidity (30). This is not well worded. As used in the paper, the statement is true of any antibody or other large multi-valent molecule.

6. KKO is supposed to be a potent activator of platelets, yet it behaved like a Group 1 antibody in these studies. Some comment about this unexpected finding would be appropriate.

Reviewer #2 (expert in platelets biology)
Remarks to the Author:

Summary of the key results
Originality and interest

In the current manuscript Nguyen et al analyse the differences in human PF4-binding antibodies, which can be divided in group 1 (antibodies that bind to PF4/polyanion complexes that do not activate platelets), group 2 (antibodies that bind to PF4/polyanion complexes that do activate platelets, and cause HIT) and group 3 (antibodies that bind to PF4 without the need for polyanions, and that do activate platelets, and cause "autoimmune HIT"). Group 1 abs have low binding forces to PF4/P, group 2 abs intermediate, whereas group 3 abs have high binding forces and, like group 2, display a biphasic binding curve. High affinity group 3 antibodies bind directly to PF4, cause clustering (as do the polyanions) and by this expose binding sites for e.g. group 2 abs.

This not only provides strong evidence for the working mechanism of group 3 autoantibodies, but might also be important for other autoimmune disorders, where a first group of antibodies might induce conformational changes in the antigens that next provoke the generation of disease causing anti-neo-epitope abs.

Data & methodology

Excellent quality of data and presentation with different experimental setups that corroborate each other.

One criticism clearly is the rather limited number of patients studied: group 1 = 3, group 2 = 3, group 3 = 4. However I can imagine that especially group 3 patients are relatively rare (?).

Suggested improvements

Major

- Can group 3 abs be isolated by using a PF4 column instead of a PF4/H column, after which the non-PF4-binders might be isolated by a PF4/H column? Does this then recover the high affinity and clustering ones and the low affinity ones respectively? Are the PF4 binders still able to activate platelets (they probably should not, according to the two step hypothesis?), if not, can platelet

activation be restored by adding the PF4/H binders on top of the PF4 binders?

- Along the same lines: PF4 clustered by high affinity group 3 abs can recruit to different extents group 2 abs, group 1 and KKO (fig 6d): the experiment should be completed by checking whether also additional group 3 abs can be recruited, as this is what is proposed to be happening in patients of group 3.

Minor

- The nomenclature used in the field is at least somewhat confusing: patients with group 3 abs, that do not require heparin to have a PF4-dependent platelet activation, are named autoimmune HIT, although obviously by definition, there is no "heparin-induced" thrombocytopenia.

Equally the same holds true for classifying group 3 abs as anti-PF4/P-ABS as there is no need for "P" here.

- Intro, line 86: "little is known about the features of anti-PF4/P-ABS which determine their biological effects": likely something should be said about the role of FcγRIIA? Further, a statement would be in place to the correlation between a positive HIPA or SRA and actual clinical HIT.

- Results, line 114-115: the lower density/.../ for the purified group-1 ABS are likely caused/...to/ some loss during the purification process: however the same concentrations (20 µg/ml) of the different groups ABS were used in the experiment. So the argument for the lower density due to AB loss seems odd?

- Discussion, line 272: ABS /.../> 100 pN bind to PF4 and activate platelets: the latter has not been demonstrated: they cluster PF4, however it is not clear whether they also are the activators, or whether the lower affinity ABS in group 3 are actually binding upon clustering and then activating.

- Fig6: in top c: please add PF4/H, in d: statistics needed.

- S table 1: legend needed

Clarity and context: lucidity of abstract/summary, appropriateness of abstract, introduction and conclusions

Only after going through the complete manuscript, the link between the measurement of the binding forces and the proposed mechanism of action of the group 3 ABS, became clear. In the abstract the link between the different forces and line 40 "these anti-PF4-autoantibodies substitute for heparin" was enigmatic. Maybe this can be clarified by referring to the capacity of these antibodies, as they bind so strongly, to bring the PF4-molecules together, thus able to substitute for heparin etc..

Reviewer #3 (expert in SMFS)

Remarks to the Author:

Complex formation of anti-PF4/P-ABS with PF4 has been shown to be important in bacterial opsonization, HIT, and autoimmune-HIT. The authors attempted on a biophysical characterization of this interaction using mainly SMFS and ITC, and put a lot of effort into their studies. Unfortunately, although they presented a huge amount of data, they failed to arrive at a comprehensive and clear picture.

The conditions for the SMFS studies are very unusual. The large indentation force of 300 pN and the

long resting time of 1 s may very likely cause unspecific adhesion. Most importantly, the authors did not show any blocking experiments, which clearly is standard in the field. Therefore, it is questionable whether the forces measured in this study can be attributed to specific antibody/antigen interactions at all. This is supported by the fact the variation of the forces is unusually large. Even if the authors attribute this variation to the heterogeneity of their antibodies, they also state in the manuscript that they are not sure, whether they measured monovalent and bivalent interaction. In addition, the SMFS data are not consistent, e.g. in figure 4 the force distributions should be equal when the configuration of tip- and surface-bound molecules is flipped. This reviewer acknowledges that the authors also investigated the loading rate dependence, however, which force data were selected here? The scattering in figure S3 appears to be much less than in all other SMFS figures. Taking this into account, how reliable are the k_{off} values? The authors also did not acknowledge the work of the pioneers and only cited their own studies.

The presentation of the ITC data is somewhat confusing. The figures in the main manuscript contain only incomplete data, whereas full tables are shown in the supplementary information. From where do the data in the tables come from?

It appears that the SMFS data are not fully consistent with the ITC data. Overall, it is more a patchwork of data that is shown and compared and not the full set of parameters that can be extracted from the two methods.

The authors claim to have discovered a new mechanism for antibody mediated autoimmune disease, in that group-3 ABS binding changes PF4 and induces expression of a neoantigen, similar (or even equal) as heparin does. To the perspective of this reviewer, this is largely speculative and not unequivocally supported by the data. Similar forces measured in SMFS and/or small enthalpy changes measured in ITC do not appear sufficient to draw such a conclusion.

Reviewers' comments:

Reviewer #1 (expert in HIT)

Remarks to the Author:

Studies were done to define how functional/biologic properties of antibodies produced by patients treated with heparin correlate with clinical manifestations, ranging from no symptoms to severe thrombocytopenia. Patient antibodies (ABS) studied were broken down into three groups. All three types of antibody react with heparin/platelet factor 4 complexes. Group-1 ABS are negative in standard platelet activation assays. Group-2 ABS activate platelets when low dose heparin is present. Group-3 ABS activate platelets in the absence of heparin and are thought to cause more severe disease (thrombocytopenia, thrombosis) than the other types, even in patients not given heparin. Findings made are interpreted as showing that behavior of the 3 types of ABS can be explained on the basis of their affinity/avidity for heparin/PF4 complexes (Group-3 > Group-2 > Group-1). This is an important finding because, although it is generally thought that Group-2 and 3 antibodies are more likely to cause disease than the much more common Group-1 antibodies, the molecular basis for this difference is not well understood. ON the basis of findings made using SMFS, the authors further suggest that Type 3 ABS bind to PF4 and modify the chemokine in such a way that it is recognized by Type 1 and Type 2 ABS even in the absence of heparin and that this leads to "recruitment" of Type 2 and Type 1 ABS into an "autoimmune process." It is proposed that findings made may have implications for other types of autoimmune disease.

Major comments

1. The authors suggest that Group-3 ABS are "autoantibodies" and that they act like heparin in the sense that they aggregate PF4 to produce complexes that are then recognized by Group-2 and perhaps Group-1 ABS, thus "recruiting" the latter ABS "...into an autoimmune process." The SMFS studies appear to be consistent with this possibility but considerable doubt can be raised as to whether single molecule observations like these can be extrapolated to what may be happening in a patient who has a "Group-3" antibody.

Response: We fully agree that SMFS experiments alone cannot reflect the *in vivo* situation. We, therefore, used the best *ex vivo* system which we currently characterize PF4/H antibodies, that is a functional assay (HIPA). It is well accepted in the literature that the findings of the functional assay correlate very reasonably with the *in vivo* capacity of antibodies to induce a procoagulatory syndrome. Whether antibodies finally cause a breakthrough in a specific individual not only depends on the characteristics of the antibodies, but also on cofactors modulating platelet activation, thrombin generation, and finally thrombus formation.

As shown by the HIPA tests with human sera (group-1, group-2, and group-3) or with the purified ABS, only the group-3 ABS strongly activate platelets in the absence of heparin. All experiments were performed with the sera and the purified PF4/H ABS from the same patients.

Even more importantly, we selected the patients based on the clinical presentation. Thus, the group-3 ABS were isolated from patients who presented with spontaneous HIT and developed autoimmune HIT *in vivo*.

We then established that their serum activates platelets in the absence of heparin, then showed that their affinity purified PF4/H-ABS activate platelets in the absence of heparin, and then demonstrated that these antibodies show the typical group-3 behavior in SMFS.

We only found antibodies which activated platelets independently of heparin in those patients who presented clinically autoimmune HIT.

For obvious reasons, we cannot reinject group-3 ABS into humans. Nevertheless, we strongly feel that we provide a clear line of evidence that these group-3 ABS are associated with the clinical presentation of autoimmune HIT. We also clearly show that these antibodies activate platelets independently of heparin even at relatively low concentrations. We cannot prove that the group-3 ABS also cluster PF4 *in vivo* and recruit group-2 ABS. This is now addressed in the discussion.

There are various ways to track formation of large IgG/PF4 aggregates in solution (e.g., light scattering). Why don't the authors incubate their purified HIT antibodies with PF4 in solution at various concentrations and see whether larger aggregates actually do form in the absence of heparin when Group-3, but not Group-2 or Group-1 ABS are present? They could even test directly whether Group-3 ABS recruit Group-2 ABS into the complexes.

Response: We thank the Reviewer for this very helpful suggestion. The suggested experiments fully confirm our hypothesis. We performed the suggested DLS experiments (Fig. 1 below).

Figure 1. Group-3 ABS cluster PF4 allowing other antibody groups to bind as obtained by dynamic light scattering (DLS). (A) Group-3 ABS /10 μg/ml formed unstable complexes with PF4 as indicated by large variation at PF4 concentrations <4 μg/ml, stable complexes from 3 to 6 μg/ml and larger complexes at ≥7μg/ml. A PF4 concentration of 5 μg/ml was selected to incubate with different antibody groups. (B) PF4 and group-3 ABS alone show sizes smaller than 10 nm which are much smaller than the sizes of the positive control PF4/heparin complexes (grey). Complexes formed when control IgG, group-1, group-2, or group-3 ABS were incubated with PF4 in the absence of heparin (blue, B). (C) The sizes of PF4/group-3 complexes increased in the presence of group-2 ABS. This demonstrates that group-2 ABS recognize PF4 bound to group-3 ABS.

When group-3 ABS 10μg/ml were incubated with PF4 in concentrations ranging from zero to 11 μg/ml, PF4/group-3 complexes increased in size more than 10-fold after 30 min incubation (Fig. 1A). At PF4 concentration <4 μg/ml, PF4/group-3 complexes are unstable as indicated by the large variation of sizes, stable at the concentration between 4 and 7μg/ml as indicated by small variation, and significant larger complexes were formed at ≥7μg/ml. We, therefore, selected a PF4 concentration of 5μg/ml to investigate binding with other antibody groups, while IgG concentrations were kept at 10 μg/ml. PF4 and group-3 ABS (black) alone show sizes smaller than 10 nm (Fig. 1B below). PF4/H complexes were used as control (grey, Fig. 1B below). Control IgG and group-1 ABS did not form complexes with PF4, group-2 ABS showed minimal complex formation, while large complexes were formed between PF4 and group-3 ABS (blue, Fig. 1B below). The sizes of PF4/group-3 complexes are even larger than that of PF4/H complexes. We then incubated the PF4/group-3 complexes with the other antibody groups for 30 min before DLS measurements. The results show that control IgG and

group-1 ABS do not significantly increase the sizes of PF4/group-3 ABS complexes, but group-2 induced larger aggregates (blue, Fig. 1C). This confirms that group-2 ABS bind to PF4/group-3 complexes.

We now added the DLS data to Fig. 5g, Fig. 6e and Supplementary Fig. 6B of the manuscript.

Alternatively, they could plate a Group-3 ab, allow it to capture PF4, see whether this creates a target recognized by Group-2 and Group-1 ABS. Positive findings would greatly strengthen claims made on the basis of the SMFS findings.

Response: With all respect for the Reviewer, the recommended experiment has actually performed as shown in Fig. 6 (main text). There, we immobilized group-3 ABS on AFM cantilevers, pre-selected the cantilevers with ABS showing strong binding forces with PF4/H complexes and let them capture PF4 to form a PF4/group-3 complex. Then, the interactions between PF4/group-3 complexes on the cantilevers and the immobilized group-2, group-1 ABS or KKO plated on the substrates were measured. Comparing with PF4/H complexes, PF4/group-3 complexes show similar interaction forces and rupture force distributions (Fig 6b-d, main text). We, therefore, conclude that the group-3 ABS capture PF4 and created a target recognized by Group-2 and Group-1 ABS.

We plated the ABS on the AFM cantilever instead of coating the entire antibody fractions of group-3 ABS on the substrate to overcome the fundamental problem of the heterogeneity of antibodies. If we coat the entire group-3 antibody fractions on the substrate, which does contain all the three antibody types (weak, strong, and super strong ABS (Fig. 4, main text)), we would have an uncontrolled mixture on the plate with variable results of the antibodies bound to the cantilever because we can never differentiate whether the antibody on the cantilever is interacting with a coated low affinity, intermediate affinity or high affinity antibody. If we would increase the concentration of IgG bound to the substrate to increase the absolute number of the super strong ABS, high surface density will induce artifacts in SMFS such as multiple rupture events which may result in inaccurate results.

Unfortunately, we cannot use an EIA, as both, the group-3 ABS as well as the group-2 or group-1 ABS are all human IgG and cannot be differentiated by our secondary antibody.

2. Line 204: "Thus, both reverse and non-reverse experiments consistently show that the original sera likely contain polyclonal ABS with different binding affinities to PF4/H complexes and indicate that the capacity of anti-PF4/P- ABS to activate platelets depends on their binding strength to PF4/H complexes."

The authors' data convincingly show that Group-3 ABS recognize PF4/P complexes with higher affinity than Group-2 or 1 ABS and that the former ABS activate normal platelets even in the absence of low dose heparin. These findings are important, but it doesn't necessarily follow that the correlations reflect a cause-and-effect relationship. The authors' conclusions would be strengthened if they could identify what is being recognized on the platelet surface by Group-3 ABS in the absence of heparin and by the Group-2 ABS in the presence of heparin in the HIPA studies (Fig. 2). This question is very important. Surely it at least deserves some discussion.

Response: The Reviewer raises an important point which we are currently addressing in a new series of experiments in which we are not only working on isolated/purified PF4/H antigens immobilized on a solid substrate, but also assessing the interaction of the antibodies with PF4 and PF4/H complexes on cell surfaces. It has been proposed (Padmanabhan et al. Blood 125 (2015) 155) that chondroitin sulfate is likely the binding partner for PF4 on the platelet surface allowing PF4/H ABS to bind in the same way they do with PF4/H complexes

(Fig. 2A-B below). Therefore, we performed the HIPA test with chondroitinase coated platelets. The results showed a delay in platelet activation by group-3 ABS (Supplementary Fig. 2), while group-2 ABS still did not activate platelets. This indicates that chondroitin sulfate is at least partly involved in PF4 binding to the platelet surface, consistent with the previous study. In addition, we performed a pilot of experiments measuring binding of the isolated antibodies to the platelet surface and found no change in the rupture force in the presence and absence of chondroitinase for group-2 ABS comparable to control IgG (Fig. 2C-D below), while group-3 ABS show a significant reduction of the interaction force when platelets are incubated with chondroitinase (Fig. 2E below). However, the interaction was not completely inhibited. The experiments using live platelets are highly complex and it will take considerable additional time to run all necessary controls, which is beyond the scope of the present manuscript.

Figure 2. Group-2 and group-3 ABS recognize PF4 on the platelet surface. (A-B) Proposed model of PF4-dependent, heparin-independent HIT antibody-mediated platelet binding/activation by Padmanabhan et al. (A) HIT antibodies bind to PF4/H complexes formed on or near the platelet surface, and then cross-link platelet FcγRIIa, triggering platelet activation. (B) An alternative model suggested that platelet-activating antibodies are able to recognize subtle conformational changes induced in PF4 when it binds to chondroitin sulfate displayed on the platelet membrane. Our preliminary SMFS results show that rupture forces between washed platelets and group-2 ABS (D) are somewhat higher than those of control IgG (C) but do not change in the presence of chondroitinase (gray), while binding forces of group-3 ABS were much higher for group 3 ABS and significantly decreased in the presence of chondroitinase. This strongly indicates that group-3 ABS recognize PF4 bound to chondroitin sulfate, but probably additional binding partners are involved as not all binding forces are inhibited.

Our results indicate that the group-3 ABS might bind to PF4 bound to chondroitin sulfate (GAG) while group-2 ABS did not. The group-2 ABS require a higher density of PF4 on the platelet surface in order to bind as we observed a strong interaction force when platelets were coated with higher concentrations of PF4 (data not show). Small quantities of PF4 may always be present on the platelet surface in a complex with GAG and can be recognized by the group-3ABS. If bound IgG is clustered sufficiently, IgG Fc domains cross-link FcγRIIa, leading to platelet activation. When some platelets are activated, they release more PF4 and

can further increase levels of GAG/PF4 on the platelet surface, leading to additional antibody binding and acceleration of the activation process.

Furthermore, we have recently shown (Brandt et al. 2014 and Brandt et al. 2015) that platelet polyphosphates are also binding partners for PF4 and it is likely that both chondroitin sulfate and polyphosphates, and potentially also other polyanions on the platelet surface may function as binding partners for PF4. As suggested by the Reviewer, we now expand the discussion on how group-3 ABS bind to platelets in the absence of heparin.

3. Line 292: "Previously, we have shown that polyanions of certain length and charge density induce the conformational change in PF4 (5, 28), required for binding of anti-PF4/P-ABS, by forcing two PF4 together (16). This results in fusion of the positive charge cloud of two PF4 tetramers and the release of additional energy, required to switch the structure of the PF4 protein for expression of the neo-epitope (16).

References 5 and 28 do show that various polyanionic compounds induce structural changes in PF4 but did not directly demonstrate any connection between these changes and HIT antibody binding. In Ref 16, it was shown that PF4 molecules repel each other because of their strong positive charges but that when PF4 is treated with glutaraldehyde to cross-link PF4 monomers "internally," weak PF4-PF4 interaction can be demonstrated by SMFS. The authors suggest this finding supports the view that PF4-PF4 bonds exist in clusters of PF4 aggregated by "long" heparin molecules. In the view of this reviewer, none of these data support the authors' contention (expressed in lines 292-296 and in many other places in the text) that Group-3 HIT ABS can induce changes in PF4 that lead to ".....expression of the neoantigen" recognized by Group-1 and Group-2 ABS and in this sense can "substitute for heparin." The SMFS studies are consistent with this possibility but it is a real stretch to suggest that these findings can be used to draw conclusions about what happens in a patient.

Response: We now changed the discussion and references accordingly.

In addition to the ITC, EIA and SMFS results shown in Fig. 5 in the previous version of the manuscript, we have now performed two additional experiments to i) prove that group-3 ABS bind to PF4 alone by purifying them *via* a PF4-column (Fig.5h), and ii) prove that the PF4/group-3 complexes allow binding of group-2 ABS as shown by DLS experiments (Fig.6e) and may be also group-1 ABS. Therefore, we are now confident to state that group-3 ABS bind to PF4 and that these complexes are recognized by group-2 ABS and in this sense can "substitute for heparin." As group-2 ABS recognize specifically a neoepitope on PF4, binding of group-2 ABS clearly confirms that PF4 must have undergone a conformational change after binding to group-3 ABS.

4. The EM studies of the HIPA reactions shown in Fig. 2 are impressive. Were these studies done with patient serum or with purified antibodies? Were any HIPA studies done with the purified antibodies to confirm that their reactions mirror those obtained with the original serum?

Response: The EM studies were done with the affinity purified antibody fractions and the purified antibody fractions behaved identically to the serum of the patient in the HIPA test. We now make it more clearly in the legend of Figure 2 that these data were obtained with the affinity purified antibodies.

5. Supplementary Fig 1: It is unusual to be able to elute antigen-specific IgG from serum in quantities sufficient to produce the bands shown in Suppl Fig 1. Were any studies done to determine experimentally what fraction of IgG present in the bands shown in lanes 9-11 is actually antibody specific for PF4/P? The gels could, for example, be performed before and after absorbing the eluates with beads coated with PF4/H complexes.

Response:

- For all purifications, the concentration of IgG in the eluate was very low. Therefore, we concentrated the eluates using the standard centrifugation method with a size exclusion filter of 100 kDa and used the same concentration (100 µg/ml) for the SDS PAGE in Supplementary Fig. 1. For titrating the antibodies in the EIA and the HIPA tests, the concentrations were raised up to 150 µg/ml (Fig. 1-2). We have now added the concentration of ABS after concentrating to the legend of the Supplementary Fig. 1.

- In regards to the suggestion of the Reviewer to pre-adsorb the eluate with PF4/H complexes, there must be a misunderstanding. We purified the antibodies by two-steps, that are, total IgG *via* protein G column and then PF4/H ABS *via* PF4/H column. We then tested the eluates of the PF4/heparin column for contaminating PF4. If a PF4 band appeared on the SDS PAGE, the antibody fraction was passed again through a protein G column to remove PF4 and the PF4/H ABS were again eluted and retested for PF4 contamination.

6. The discussion is long and rambling and contains many seemingly disconnected paragraphs. An example is the last 4 paragraphs running from lines 296-317. The most interesting and important finding described in this paper is the convincing correlation between the functional/biological properties of the different HIT antibody groups and their affinity/avidity for heparin/PF4 complexes. The paper could be greatly improved by focusing on this and a few other significant findings that are well-documented and by avoiding undue speculation about the significance of the SMFS findings for HIT pathogenesis and for autoimmune diseases in general.

Response: We condensed the discussion as suggested by the Reviewer. The differences among antibody groups are now discussed in the first part of the discussion; the 'Technical issues' are addressed in the second part, while the more speculative parts are discussed in the "Outlook" section.

Minor comments:

1. Lines 35-36. "HIT occurs when anti-PF4/P-antibodies activate PF4/H-coated platelets; while..." This reviewer is unaware of any published evidence showing that HIT is caused by such a process.

Response: We have shown this experimentally many years ago, when we incubated platelets with highly sulfated polyanions, washed them and then added HIT antibodies:

Greinacher A, Michels I, Liebenhoff U, Presek P, Mueller-Eckhardt C. Heparin-associated thrombocytopenia: immune complexes are attached to the platelet membrane by the negative charge of highly sulfated oligosaccharides. *Br J Haematol.* 1993 Aug;84(4):711-6. PMID: 8217831.

2. Lines 55-56. "Binding of PF4 to polyanions results in its conformational change (5, 6) exposing the binding site(s) for anti-PF4/P-ABS. Reference 6 is inappropriate because the only antibody studied was KKO, which binds to PF4 in the absence of heparin, and the only polyanion studied was fondiparin, which induces little or no conformational change in PF4.

Response: We removed reference 6, which was originally included because it proposes the model of conformational changes in PF4 when heparin binds.

3. Lines 267-269. "We show for the immune response against PF4/P complexes that the biological activity of anti-PF4/P-ABS depends on their binding affinity, as measured by binding forces of highly purified anti-PF4/P-ABS." It would be more accurate to say that

biologic activity "correlates closely" with binding affinity since "cause-and effect" was not documented. And the term "immune response" is inappropriate because the paper does not deal with this subject (induction of an HIT antibody response).

Response: Thank you for the clarification, we fully agree and changed the manuscript accordingly.

4. Lines 310-311. "In nearly all antibody-mediated autoimmune diseases, the antibody specificities found in severely affected patients are also found in asymptomatic individuals of the general population." Can the authors provide a reference for this generalization?

Response: Both references below address this issue and are now included into the manuscript.

R Hal Scofield, Autoantibodies as predictors of disease, Lancet 2004; 363: 1544

Leslie D, et al. J Clinical Investigation 108 (2001) 1417: "The most common autoantibodies, namely rheumatoid factor (RF), an autoantibody to the Fc portion of IgG, and anti-nuclear antibodies (ANAs) are frequently found in rheumatoid arthritis (<75%) and systemic lupus erythematosus (SLE) (90–100%), respectively. However, these autoantibodies are also found in 5–10% of normal individuals, especially the elderly. For this reason, the number of normal individuals with these autoantibodies exceeds the number of patients with these autoantibodies by many-fold."

5. Lines 320-322. "It has been shown for other prothrombotic autoantibodies, e.g. against $\beta 2$ Glycoprotein 1 in patients with the antiphospholipid syndrome that bivalent binding of autoantibodies largely increases their binding avidity (30). This is not well worded. As used in the paper, the statement is true of any antibody or other large multi-valent molecule.

Response: We changed the wording, what we want to say is that if an antibody can bind with both arms it binds stronger when compared to the situation where it can only bind with one arm.

6. KKO is supposed to be a potent activator of platelets, yet it behaved like a Group-1 antibody in these studies. Some comment about this unexpected finding would be appropriate.

Response: We address this in the discussion. This was also a surprising finding for us. In both SMFS and ITC, we did not observe strong interactions of KKO with PF4/H complexes, although it activated platelets coated with PF4 in the HIPA test. One hypothesis is that the antibodies interact less well with PF4/heparin complexes under the experimental conditions we used. In addition, KKO bound with weak binding forces but shows lower k_{off} values than group-1 ABS, which are comparable with that of group-2 ABS, indicating higher affinity. We address both issues now in the Discussion part and state that the lower k_{off} value is likely associated with a longer persistence of the antibody on the platelet surface, which is long enough to cross-link enough Fc receptors to induce platelet activation.

Reviewer #2 (expert in platelets biology)

Remarks to the Author:

Summary of the key results

Originality and interest

In the current manuscript Nguyen et al analyze the differences in human PF4-binding antibodies, which can be divided in group-1 (antibodies that bind to PF4/polyanion complexes that do not activate platelets), group-2 (antibodies that bind to PF4/polyanion complexes that do activate platelets, and cause HIT) and group-3 ABS (antibodies that bind to PF4 without the need for polyanions, and that do activate platelets, and cause "autoimmune HIT"). Group-1 ABS have low binding forces to PF4/P, group-2 ABS intermediate, whereas group-3 ABS have high binding forces and, like group-2, display a biphasic binding curve. High affinity group-3 ABS bind directly to PF4, cause clustering (as do the polyanions) and by this expose binding sites for e.g. group-2 ABS.

This not only provides strong evidence for the working mechanism of group-3 autoantibodies, but might also be important for other autoimmune disorders, where a first group of antibodies might induce conformational changes in the antigens that next provoke the generation of disease causing anti-neo-epitope ABS.

Data & methodology

Excellent quality of data and presentation with different experimental setups that corroborate each other.

One criticism clearly is the rather limited number of patients studied: group-1 = 3, group-2 = 3, group-3 = 4. However, I can imagine that especially group-3 patients are relatively rare (?).

Response: We thank the Reviewer for the very positive judgment of our study. We raised the numbers to 5 sera for each group for EIA (Fig. 1), HIPA (Fig. 2) and SMFS (Fig. 3 and 4) experiments. Group-3 patients are really rare and we need a lot of plasma/serum of those patients to purify large enough quantities of antibodies due to the multi-step purification procedure and the large amount of antibodies needed for the control experiments showing that the purified antibody fraction behaves like the original serum. It took us years to collect the samples to perform the present study.

Suggested improvements

Major

Can group-3 ABS be isolated by using a PF4 column instead of a PF4/H column, after which the non-PF4-binders might be isolated by a PF4/H column? Does this then recover the high affinity and clustering ones and the low affinity ones respectively? Are the PF4 binders still able to activate platelets (they probably should not, according to the two-step hypothesis?), if not, can platelet activation be restored by adding the PF4/H binders on top of the PF4 binders?

Response: That is an excellent suggestion and we have now purified PF4-ABS using PF4 coated beads and tested them. The eluate shows that depending on the characteristics of sera the concentrations of eluted antibodies using the PF4-column are different, i.e., 8-, 17-, 40-

µg/ml for group-1, group-2, and group-3, respectively. We concentrate the eluates to the concentration of 50 µg/ml and tested them in the HIPA. The results showed that the PF4-ABS from group-3 sera activate platelets in the absence of heparin at an earliest lag-time phase (~5min), while the antibodies purified by a PF4 column of the other antibody groups do not activate platelets neither in the presence, nor in the absence of heparin. The non-PF4-binders were then passed through a PF4/H column to collect PF4/H ABS. These PF4/H ABS from group-2 and group-3 sera (concentration 50 µg/ml) still activated platelets, but only in the presence of heparin, while control IgG and group-1 ABS did not.

Along the same lines: PF4 clustered by high affinity group-3 ABS can recruit to different extents group-2 ABS, group-1 and KKO (fig 6d): the experiment should be completed by checking whether also additional group-3 ABS can be recruited, as this is what is proposed to be happening in patients of group-3.

Response: We are certain that group-3 ABS will be recruited by PF4/group-3 complexes. However, having antibodies with the same binding affinity on the cantilever and on the solid phase has the risk that PF4 will be ruptured from the PF4/group-3 complexes on the cantilever. In other words, the PF4/group-3 complexes will be damaged after some measurements. Therefore, we did not carry out this experiment. In contrast to group-3 ABS, other ABS (group-1 and group-2) interact weaker and they cannot disrupt PF4 from PF4/group-3 complexes.

Minor

- The nomenclature used in the field is at least somewhat confusing: patients with group-3 ABS, that do not require heparin to have a PF4-dependent platelet activation, are named autoimmune HIT, although obviously by definition, there is no "heparin-induced" thrombocytopenia.

Equally the same holds true for classifying group-3 ABS as anti-PF4/P-ABS as there is no need for "P" here.

Response: We fully agree with the Reviewer that the nomenclature is somewhat confusing. To be consistent with the literature, we followed the nomenclature recommended by Warkentin et al. in BLOOD (Warkentin TE et al. Spontaneous heparin-induced thrombocytopenia syndrome: 2 new cases and a proposal for defining this disorder (*Blood* 2014, **123**(23): 3651). We defined all ABS that bind in PF4/H EIA as PF4/H-ABS. For the individual cases, we named them group-1, group-2 or group-3 ABS.

- Intro, line 86: "little is known about the features of anti-PF4/P-ABS which determine their biological effects": likely something should be said about the role of FcγRIIA? Further, a statement would be in place to the correlation between a positive HIPA or SRA and actual clinical HIT.

Response: Thank you very much for helping us to improve the introduction. The introduction has been changed accordingly and we added now two references (Rollin et al, *Blood* 2015, **125**: 2397; Rollin et al, *Thromb Haemost* 2016, **116**: 799).

- Results, line 114-115: the lower density/.../ for the purified group-1 ABS are likely caused/...to/ some loss during the purification process: however the same concentrations (20 µg/ml) of the different groups ABS were used in the experiment. So the argument for the lower density due to AB loss seems odd?

Response: The Reviewer is correct. This was an error on our side. What the experiment indicates is that it might be that the group-1 ABS had been present at a higher concentration in

the original serum. We changed the manuscript accordingly.

- Discussion, line 272: ABS \geq 100 pN bind to PF4 and activate platelets: the latter has not been demonstrated: they cluster PF4, however it is not clear whether they also are the activators, or whether the lower affinity ABS in group-3 are actually binding upon clustering and then activating.

Response: Thank you very much for the comment. The corrected wording is “while antibodies with binding forces >100 pN activate platelets even in the absence of polyanions”. In addition, as the Reviewer suggested, we have purified PF4 ABS by PF4-column. These PF4 ABS activated platelets in our HIPA test rapidly, and their interaction forces with PF4/H are ≥ 100 pN (Supplementary Fig 7A). They also formed complexes with PF4 as shown in our DLS experiments. The manuscript has been changed accordingly.

- Fig6: in top c: please add PF4/H, in d: statistics needed.

Response: Fig. 6c-d has been corrected.

- S table 1: legend needed

Response: Legend has been added to Table S1.

Clarity and context: lucidity of abstract/summary, appropriateness of abstract, introduction and conclusions

Only after going through the complete manuscript, the link between the measurement of the binding forces and the proposed mechanism of action of the group-3 ABS, became clear. In the abstract the link between the different forces and line 40 "these anti-PF4-autoantibodies substitute for heparin" was enigmatic. Maybe this can be clarified by referring to the capacity of these antibodies, as they bind so strongly, to bring the PF4-molecules together, thus able to substitute for heparin etc.

Response: We changed the manuscript accordingly.

Reviewer #3 (expert in SMFS)

Remarks to the Author:

Complex formation of anti-PF4/P-ABS with PF4 has been shown to be important in bacterial opsonization, HIT, and autoimmune-HIT. The authors attempted on a biophysical characterization of this interaction using mainly SMFS and ITC, and put a lot of effort into their studies. Unfortunately, although they presented a huge amount of data, they failed to arrive at a comprehensive and clear picture.

The conditions for the SMFS studies are very unusual. The large indentation force of 300 pN and the long resting time of 1 s may very likely cause unspecific adhesion.

Response: We thank the Reviewer for this critical comment. We agree that the applied force of 300 pN can induce some indentation on biomolecules, and the resting time of 1s can increase the chance of non-specific interactions.

We used the contact time of 1s because we know from our functional assays that there is a lag-time of several minutes until PF4/H ABS induce platelet activation. The complexes between PF4 and heparin expose a complex and structurally complicated antigen. It is unknown whether refolding of the protein after antibody binding influences binding affinity. Most probably, the interaction of three partners (PF4/H/antibody) requires a longer time to bind in 3D than other systems with only two binding partners. As an example for fibrinogen- α IIb β 3 interaction, Litvinov et al (*Biophys J.* **89** (2005) 2824) proposed a model that fibrinogen interacts strongly with α IIb β 3 receptor *via* two binding steps: i) the initial interaction, and ii) the molecular reorganization. The time required for this stable interaction is 70ms. In addition, Dupres et al. found that the Heparin-binding haemagglutinin (HBHA) adhesion force and adhesion frequency increased exponentially with contact time and saturated at even much longer contact time: 2s (*Nature Methods* **2** (2005) 515). They suggested that the dependence of adhesion frequency and adhesion force on the contact time indicates that the HBHA complex is formed *via* multiple intermolecular electrostatic bridges.

We assumed that our system is even more complex than the fibrinogen- α IIb β 3 or the HBHA interaction and therefore, 1s resting time was applied. In addition, we are planning to investigate as a next step antibody interaction with PF4/H complexes on the platelet surface. Here, the interaction partners are even more complex and the likelihood that molecular rearrangements on the cell membrane influence antibody binding affinity is relatively high. Thus, the cell-based experiments will require a relatively long incubation time. To be able to compare the current experiments with later cell-based experiments makes it necessary to use the similar contact times.

Regarding the indentation force, we applied the force similar to that of the group of Prof. Yves F. Dufrene used for SMFS experiments of biomolecular interactions: e.g., Sullan, R. M. et al, *ASC Nano*, **9** (2015) 1448 or Tripathi P, *ASC Nano*, **7** (2013) 3685 or Dupres et al., *Nature Methods*, 2005, **2**, 515), in which an applied force between 250 to 500 pN was used. Other groups also used higher forces (e.g. Francius et al, *ACS Nano*, 2008, **2**, 1921; Carvalho et al., *ACS Nano*, 2010, **4**, 4609; Formosa et al, *Nanomedicine* **11** (2015) 57; Kedrov *et al*, *Journal of Structural Biology* **159** (2007) 290).

We now performed additional experiments to exclude major artifacts due to our measurement conditions (Fig. 3 below). The interactions between control IgG and PF4/H complexes were measured at lower applied force (100 pN, black) and higher applied force (300 pN, blue) with resting time of 0s and 1s for comparison. These experiments show that lower applied force

and shorter contact time reduce unspecific bindings. The counts induced by a higher applied force (300 pN) together with longer resting (1s) is only ~2% difference from the lower applied force (100 pN)-no resting time (0s). As the average of total counts obtained by control IgG was approximately in the range of 4%, we considered this as background in our study, and always subtracted this background from the force histograms of other antibodies (Fig. 3-4, main text). After subtracting, we still obtained specific bindings of ~12 % for KKO, ~10 % for group-1, ~20 % for group-2, and ~38 % for group-3ABS.

We consider that all technical artifacts have the same influence on the various antibodies and do not influence the relative difference in binding forces and binding counts between the different antibody groups.

Figure 2. Comparison between 100- and 300pN applied force at a resting time of 0- and 1s. Control IgG show ~4% counts in three independent experiments. The ‘counts’ at the 300 pN applied force (blue) is higher than that at 100 pN (black) and the difference is ~2%. With 1s resting time, the ‘counts’ increase for both 100- (red) and 300 pN (purple) applied force compared to 0s resting time (except the second experiment at 300 pN applied force, the ‘counts’ slightly increase at 0s resting time).

Most importantly, the authors did not show any blocking experiments, which clearly is standard in the field. Therefore, it is questionable whether the forces measured in this study can be attributed to specific antibody/antigen interactions at all. This is supported by the fact the variation of the forces is unusually large.

Response: We highly appreciated this comment from the Reviewer which indeed improves the quality of our manuscript. The blocking experiments are important controls to confirm that our rupture forces presented in the manuscript were due to the interaction between antibodies and their antigens.

Regarding the binding sites on PF4 to which the PF4/H antibodies bind, there is currently no way to block them specifically by any small molecule inhibitor. We tried several possibilities to block antibody binding:

1. Blocking PF4/H ABS by their antigens (PF4 or PF4/H complexes) failed because of the strong repulsive forces between PF4 or PF4/H complexes bound to the antibodies on the tips and the PF4/H complexes coated on the substrate (also observed previously; Nguyen et al, *Nanoscale* 7 (2015) 10130).
2. We therefore used a well-established method to disrupt the antigen by high heparin concentrations, which is a standard control in EIAs or functional assays used for detection of PF4/heparin antibodies in clinical assays. We coated PF4/H complexes on the substrate (Fig. 3A below) or on the AFM-tip (Fig. 3B below) and measured their interactions with

antibodies immobilized on the cantilevers or on the substrate, respectively. Then, a heparin concentration of 100 IU was added to the liquid cell and incubated for 30 min. After that, their interactions were again measured. In both cases, we found that the number of interactions (counts) drastically reduced after adding high heparin concentration (Fig. 3A-B below). After disrupting, the PF4s are coated with heparin which is the most likely explanation that also group-3 ABS showed reduced binding

Figure 3. Blocking experiments. (A) PF4/H complexes coated on the substrate are disrupted after adding high heparin concentration resulting in the drastically reductions of interaction events (counts) when group-1 or group-2 ABS interact with, while group-3 ABS still bind but much weaker. (B) The interaction reduced for all antibodies when PF4/H complexes on the tips were incubated with high heparin concentration. (C) Blocking of binding sites of PF4/group-3 ABS on the tip by group-2 ABS results in the minimal binding events.

- To block the PF4 binding site on PF4/group-3 ABS complexes, we incubated the cantilever with group-2 ABS for 30 min. After that, the cantilevers were brought to the substrate coated with either group-1 or group-2 ABS for interaction. Comparing the 'counts' before and after adding group-2 ABS, we also observed a significant reduction (Fig. 3C, above).

These blocking experiments indicate that our rupture forces presented in the manuscript were due to the interaction between antibodies and their antigens. This new data is now added to the supplementary Fig. 4.

Furthermore, to control for the true variation in binding force and artifacts, we used two controls: that are, KKO and the human control IgG. KKO is a well-characterized monoclonal antibody that mimics HIT ABS and activates platelets in the presence of PF4, while control IgG is purified from healthy donors and does not interact with PF4 or PF4/H complexes in EIA and HIPA tests. These controls do not show the large variation of binding force as group-2 and group-3 ABS. Thus, we are certain that the variability we did observe with the polyclonal group-2 and group-3 ABS is not a systematic experimental artifact, but reflects the presence of different PF4/heparin antibodies produced by different human B cells of the patient. In addition, we did show by comparing binding of the antibodies to PF4 or PF4/heparin complexes large differences in binding counts for group 1 and group 2 ABS (Fig 5f, main text), which should not be seen in the case of unspecific binding.

Even if the authors attribute this variation to the heterogeneity of their antibodies, they also state in the manuscript that they are not sure, whether they measured monovalent and bivalent interaction.

Response: We apologize for this unclear description. We surely concluded that the variation is due to the heterogeneity of our antibodies. The statement that ‘we are not sure, whether we measured monovalent and bivalent interaction’ referred to the different structures of IgG (2 Fabs per antibody) in a particular serum, e.g., among weak, strong, and super strong antibodies in group-3 ABS (Fig. 4d, main text). The binding affinity of an antibody is much higher when both Fab parts of an IgG bind to its antigen PF4/H complex). We discussed that we could not confirm whether the low binding forces of group-1 ABS (weak antibodies: binding force <60 pN) are due to monovalent interactions, while the higher forces of group-2 and group 3 ABS (strong/super strong antibodies: binding force ≥ 60 pN) are due to bivalent interactions. We have now described this clearer in the text.

In addition, the SMFS data are not consistent, e.g. in Fig. 4 the force distributions should be equal when the configuration of tip- and surface-bound molecules is flipped.

Response: We assume that the Reviewer refers to the force distributions in Fig. 3e and the ones in Fig. 4e. It can be seen that the counts and forces for the monoclonal antibody KKO and group-1 ABS are similar between the two figures. There is a slight variation of the rupture force for KKO (but mean values are still <60 pN) which may be due to the fact that KKO requires the PF4/H fixed on the substrate in order to bind. Using a long PEG linker, probably the single PF4/H complex on the tip moved freely and led to a slight reduction of binding. What differs are the binding forces and count distributions for group-2 and group-3 ABS. This is consistent with our finding that the affinity purified antibodies in group-2 and group-3 contain polyclonal antibodies with different binding characteristics. In Fig. 3 (main text) one antibody was tested with PF4/H antigens, but in the reverse experiment, the single PF4/H antigen was tested with all antibodies isolated from one patient serum. The antigen on the tip, therefore, reacted in the reverse experiment with both low- and high affinity antibodies coated on the substrate as shown by two rupture force distributions. The observed differences in the two figures are fully concordant with our concept and hypothesis.

This reviewer acknowledges that the authors also investigated the loading rate dependence, however, which force data were selected here?

Response: We thank the Reviewer for pointing out important missing information on the pulling speed for the data in Fig. 3-4. We now added the pulling speed of $1\mu\text{m/s}$ in the legend of Fig. 4.

The scattering in Fig. S3 appears to be much less than in all other SMFS figures. Taking this into account, how reliable are the k_{off} values?

Response: We thank the Reviewer for this comment. We should have described Supplementary Fig. 3 (now Supplementary Fig. 6) more clearly. The KKO and group-1 samples contain homogeneous antibodies, while group-2 and group-3 contain antibodies with different characteristics. We, therefore, pre-tested the cantilevers coated with either group-2 or group-3 ABS and selected cantilevers which had an antibody bound, which showed an interaction force of ~ 70 pN for group-2 ABS and of ~ 90 pN for group-3 ABS for the measurements in Supplementary Fig. 3. Since these antibodies were preselected, we did not observe the large variation of rupture force distributions compared to the analysis of the unselected PF4/H ABS fraction shown in Fig. 3 and Fig. 4. The data presented in Supplementary Fig. 3 were determined from the mean values of different sera. We have now added this information in the legend of the Supplementary Fig. 5.

The authors also did not acknowledge the work of the pioneers and only cited their own studies.

Response: We apologize for this. We cited mainly the references involving studies in the field of HIT. However, we are very happy to add now more references to refer to the general biophysics of antigen-antibody interactions.

The presentation of the ITC data is somewhat confusing. The figures in the main manuscript contain only incomplete data, whereas full tables are shown in the supplementary information. From where do the data in the tables come from?

Response: We apologize for an unclear description. The goal of presenting ITC data in Fig. 5 is to illustrate that the group-3 ABS interact with PF4 alone at low concentration (62.5 nM), while group-2 ABS show some interaction only at high concentration (950mM). Only group-3 ABS data could be fitted to obtain parameters shown in Supplementary Fig. 5/Panel-2. The data of group-2 ABS did not fit well with the one set of site model, and we, therefore, do not show fitting parameters.

We now also added the original ITC data of group-2 and group-3 ABS when they interact with PF4/H complexes for calculating the parameters presented in the Supplementary Figure 5/Panel-1. We now describe this more clearly.

It appears that the SMFS data are not fully consistent with the ITC data. Overall, it is more a patchwork of data that is shown and compared and not the full set of parameters that can be extracted from the two methods.

Response:

We used ITC here as a complementary method to study whether the group-3 ABS interact with PF4 or PF4/H complexes in a different way as other antibodies do. By ITC, group-3 ABS interacted strongest with PF4 alone and only group-3 ABS cluster two PF4 molecules, consistent with SMFS, PF4- and PF4/H EIA results (Fig. 5e-f), as well as the functional HIPA test (Fig. 2). We now performed two additional experiments showing that a subset of group-3 ABS could be even purified by a PF4-column (Fig. 5h), and group-3 ABS clustered PF4 to form a large complex by DLS (Fig. 5g, and answer to Reviewer 1, comment 2).

The authors claim to have discovered a new mechanism for antibody mediated autoimmune disease, in that group-3 ABS binding changes PF4 and induces expression of a neoantigen, similar (or even equal) as heparin does. To the perspective of this reviewer, this is largely speculative and not unequivocally supported by the data. Similar forces measured in SMFS

and/or small enthalpy changes measured in ITC do not appear sufficient to draw such a conclusion.

Response: We thank the Reviewer for this critical comment. As suggested by Reviewer 1 and Reviewer 2, we now performed two additional experiments as briefly described in the last comment to support the hypothesis that group-3 ABS cluster PF4 and the resulting complexes allow binding of other antibodies.

The first experiment is purification of PF4-ABS by a PF4-column, which again induced platelet aggregation in the HIPA test within 5 minutes (Fig. 5h). There is a large amount of literature that platelet activation by PF4/H antibodies requires complex formation as only immunocomplexes are able to crosslink several Fc-receptors on platelets. As our PF4-purified group-3 ABS activated platelets in the functional assay this is indirect, but very strong evidence that these antibodies cluster PF4.

The second experiment to confirm that group-3 ABS self-cluster PF4 and form complexes was performed by using dynamic light scattering (DLS) (please see response to Reviewer 1). Even more striking, the resulting complexes become even bigger when incubated with group-2 ABS directly proving that PF4 in complex with group-3 ABS exposes the binding site for group-2 ABS otherwise the group-2 ABS could not bind and increase the size of the complexes. We have now added the DLS data to Fig. 5g, Fig 6e and Supplementary Fig. 6B.

Our previous studies showed that heparin induces conformational changes in PF4 when it forms PF4/H complexes (Brandt et al, *Thromb Haemost* 2014, **112**(1): 53; Kreimann et al, *Blood* 2014, **124**(15): 2442) and that group-2 ABS only bind the PF4/H complexes and not to PF4 alone (Greinacher et al, *Thrombosis and Haemostasis* 1994, **71**:247). This we reconfirm in the present study (Fig 5e). In immunohematology, the epitopes on molecules are typically defined by antibody binding. An antibody specific for a distinct epitope can only bind if the epitope is present. By showing binding of PF4/H specific group-2 ABS to PF4/group3 antibody complexes, we directly prove that group-3 ABS induce a neoepitope on PF4.

Combining the above experiments, we provide a direct evidence that group-3 ABS binding results in changes of the PF4 conformation and expression of the antigen to which group-2 ABS bind, similar (or even equal) to what heparin does by showing direct binding in the SMFS experiments and formation of complexes by DLS experiments. Together, our results obtained by ITC, SMFS, HIPA, EIA, DLS and PF4-column purification indicated that group-3 ABS induce conformational changes in PF4 allowing binding of other heparin-dependent antibodies. We also agree that the conformational changes in PF4 induced by group-3 ABS may not be exactly the same as induced by heparin, but they clearly induce the neoepitope to which group-2 ABS bind.

Reviewers' comments:

Reviewer #1 (Remarks to the Author):

The authors have devoted a great deal of effort to the comments made and questions asked by reviewers and the MS is significantly improved. Here are a few additional remarks I believe they should consider.

1. Original comment/question. "There are various ways to track formation of large IgG/PF4 aggregates in solution (e.g., light scattering). Why don't the authors incubate their purified HIT antibodies with PF4 in solution at various concentrations and see whether larger aggregates actually do form in the absence of heparin when Group-3, but not Group-2 or Group-1 ABS are present? They could even test directly whether Group-3 ABS recruit Group-2 ABS into the complexes."

New comment: The authors performed some of the suggested studies and obtained data supporting the idea that Group 3 abs aggregate PF4. However, Fig 1C. purports to show that Group 3 abs plus PF4 recruit Group 2 abs into the complex and increase their size. Looking in the slight increase in median size and the large SD value, it appears that this is unlikely. The authors shouldn't use this Figure in the Supplementary Information unless they can document that the difference shown is statistically significant.

2. Original comment/question. "Alternatively, they could plate a Group-3 ab, allow it to capture PF4, see whether this creates a target recognized by Group-2 and Group-1 ABS. Positive findings would greatly strengthen claims made on the basis of the SMFS findings."

New comment: There was a miscommunication. The comment was intended to suggest studies that could be done in a solid phase serologic assay using an ELISA or a fluorescent end point. Since the authors have purified antibodies belonging to Groups 1-3, these can be labeled (with different fluorors for example) and detected individually.

3. Original comment/question. "2. Line 204: "Thus, both reverse and non-reverse experiments consistently show that the original sera likely contain polyclonal ABS with different binding affinities to PF4/H complexes and indicate that the capacity of anti-PF4/P- ABS to activate platelets depends on their binding strength to PF4/H complexes."

The authors' data convincingly show that Group-3 ABS recognize PF4/P complexes with higher affinity than Group-2 or 1 ABS and that the former ABS activate normal platelets even in the absence of low dose heparin. These findings are important, but it doesn't necessarily follow that the correlations reflect a cause-and-effect relationship. The authors' conclusions would be strengthened if they could identify what is being recognized on the platelet surface by Group-3 ABS in the absence of heparin and by the Group-2 ABS in the presence of heparin in the HIPA studies (Fig. 2). This question is very important. Surely it at least deserves some discussion."

New comment: In their response, the authors appear to be assuming that treating platelets with chondroitinase ABC removes all the chondroitin sulfate. In fact, such treatment removes less than half of the CS and about 24 hours is required to achieve that. Even if their treatment did remove all the CS, the findings described don't shed light on what the platelet target for Group 3 antibodies might be. And it is hard to imagine that meaningful atomic force studies can be done on platelets subjected first to chondroitinase treatment. It is not surprising that polyphosphates bind to PF4 but that seems irrelevant to the issue at hand.

4. Original comment/question Line 292: "Previously, we have shown that polyanions of certain length

and charge density induce the conformational change in PF4 (5, 28), required for binding of anti-PF4/P-ABS, by forcing two PF4 together (16). This results in fusion of the positive charge cloud of two PF4 tetramers and the release of additional energy, required to switch the structure of the PF4 protein for expression of the neo-epitope (16).

References 5 and 28 do show that various polyanionic compounds induce structural changes in PF4 but did not directly demonstrate any connection between these changes and HIT antibody binding. In Ref 16, it was shown that PF4 molecules repel each other because of their strong positive charges but that when PF4 is treated with glutaraldehyde to cross-link PF4 monomers "internally," weak PF4-PF4 interaction can be demonstrated by SMFS. The authors suggest this finding supports the view that PF4-PF4 bonds exist in clusters of PF4 aggregated by "long" heparin molecules. In the view of this reviewer, none of these data support the authors' contention (expressed in lines 292-296 and in many other places in the text) that Group-3 HIT ABS can induce changes in PF4 that lead to ".....expression of the neoantigen" recognized by Group-1 and Group-2 ABS and in this sense can "substitute for heparin." The SMFS studies are consistent with this possibility but it is a real stretch to suggest that these findings can be used to draw conclusions about what happens in a patient.

New comment: The new data are helpful, but there is still a problem justifying the claim that the HIT "antigen" is created when 2 PF4 molecules are "pulled together." How does this gibe with the idea that polyanion-induced conformational changes in PF4 create the target recognized by HIT abs? Surely the two processes can't be structurally identical, or even similar. How do the authors reconcile this apparent conflict? Note that reviewer 2 also had a problem with the suggestion that Group 3 antibodies can "substitute" for heparin. "Substitute" just isn't the right word to use here. Why not describe the mechanics of this phenomenon and let it go at that?

5. Original comment/question Supplementary Fig 1: It is unusual to be able to elute antigen-specific IgG from serum in quantities sufficient to produce the bands shown in Suppl Fig 1. Were any studies done to determine experimentally what fraction of IgG present in the bands shown in lanes 9-11 is actually antibody specific for PF4/P? The gels could, for example, be performed before and after absorbing the eluates with beads coated with PF4/H complexes.

New comment: This response offered doesn't address the question, which was "What % of the IgG in the bands shown was actually HIT antibody?"

6. Original comment/question KKO is supposed to be a potent activator of platelets, yet it behaved like a Group-1 antibody in these studies. Some comment about this unexpected finding would be appropriate.

New comment: In their response, the authors seem to be suggesting that KKO may have a lower off rate yet also a lower affinity. The authors need to be careful about this argument. By "smaller" Koff, do they mean a smaller off rate? Antibody affinity is strictly governed by the off rate. If an antibody persists longer on its target, it will have a smaller off rate and a higher affinity. So the explanation offered for the behavior of KKO seems untenable.

Reviewer #2 (Remarks to the Author):

The authors gave a thorough response, including numerous additional experiments that indeed did strengthen their proposal on the mechanism of action of HIT antibodies.

The ultimate proof that group 3 ABS are revealing the neo-epitope allowing group 2 ABS to bind and activate platelets, and thus truly mimick heparin/polyanion in this respect, indeed would be to use

F(ab)'₂-fragments of group 3 ABS in combination with group 2 intact ABS.

This is mentioned by the authors under "Outlook" to be impossible "as the polyclonal patient serum also contains weaker reacting, polyanion-dependent (group-2) PF4/P-ABS. Their F(ab)'₂ fragments would block binding of intact polyanion-dependent anti-PF4/P-ABS."

Probably correct, however if F(ab)'₂ fragments would be made from PF4-purified group 3 ABS, the relative low number of remaining group 2 F(ab)'₂ likely would no longer be a problem.

Reviewer #3 (Remarks to the Author):

The manuscript has significantly improved, yet the force spectroscopy part (Fig. S6) is still completely unclear. Which forces from the broad force distributions have been taken for Fig. S6. Indicate widths of distributions in Fig.S6, error bars, quality of fits etc.

Reviewers' comments:

Reviewer #1 (Remarks to the Author):

The authors have devoted a great deal of effort to the comments made and questions asked by reviewers and the MS is significantly improved. Here are a few additional remarks I believe they should consider.

1. Original comment/question. "There are various ways to track formation of large IgG/PF4 aggregates in solution (e.g., light scattering). Why don't the authors incubate their purified HIT antibodies with PF4 in solution at various concentrations and see whether larger aggregates actually do form in the absence of heparin when Group-3, but not Group-2 or Group-1 ABS are present? They could even test directly whether Group-3 ABS recruit Group-2 ABS into the complexes."

New comment: The authors performed some of the suggested studies and obtained data supporting the idea that Group 3 abs aggregate PF4. However, Fig 1C. purports to show that Group 3 abs plus PF4 recruit Group 2 abs into the complex and increase their size. Looking in the slight increase in median size and the large SD value, it appears that this is unlikely. The authors shouldn't use this Figure in the Supplementary Information unless they can document that the difference shown is statistically significant.

Response: The large SD was due to averaging of all measurements of several independent experiments. We now show the changes in complex size of all individual measurements using different ABS purified from different sera. This shows the increase in size when PF4/Group-3 ABS recruit Group-2 ABS much clearer (now shown in Fig. 6e). We also performed the statistics to compare the sizes of PF4/group-3 ABS complexes and their size in the presence of group-2 ABS. ANOVA test shows that the sizes are significantly different ($P=0.047$, ANOVA tests).

2. Original comment/question. "Alternatively, they could plate a Group-3 ab, allow it to capture PF4, see whether this creates a target recognized by Group-2 and Group-1 ABS. Positive findings would greatly strengthen claims made on the basis of the SMFS findings."

New comment: There was a miscommunication. The comment was intended to suggest studies that could be done in a solid phase serologic assay using an ELISA or a fluorescent

end point. Since the authors have purified antibodies belonging to Groups 1-3, these can be labeled (with different fluors for example) and detected individually.

Response: We apologize for this misunderstanding. Regarding antibody labeling, we have a problem of materials since direct labeling of antibodies causes substantial antibody loss. In addition, the labeling procedure is not specific for the Fc part and direct labeling always has a high risk to reduce immunoreactivity of the antibody by an interference of the label with the antigen binding site. This is especially problematic given the complex antigen on PF4 and the potential issue of a sterical hindrance.

Instead of labeling, we used an alternative approach in line of the suggestion of the Reviewer. We covalently immobilized group-3 ABS on the substrate *via* Au-thiol bond and amide coupling using PEG linkers, as used for immobilization of ABS on the AFM tip or on the substrate. The long PEG linker of ~30 nm allows high flexibility of group-3 ABS to avoid blocking PF4 binding site and keeps PF4/group-3 ABS complexes away from the surface, which reduces artifacts. After that, control IgG, group-1, group-2 or group-3 ABS were incubated with PF4/group-3 ABS on the substrates. After 2h, concentrations of the supernatants were determined. We found that the concentration of control IgG did not change, but those of group-1, group-2 and group-3 ABS (Fig. 1 below). The results indicated that these ABS bound to PF4/group-3 ABS complexes on the substrates. The trend of binding increases from group-1, to group-2 and to group-3 ABS which is similar to their reactions with PF4/H complexes in EIA shown in Fig.1 main text.

New supplementary figure 6B: PF4/Group-3 ABS complexes immobilized on the substrate created a target recognized by Groups 1-3 ABS. Different ABS (n=2 for each group ABS) at 25 µg/ml (black) were incubated with PF4/Group-3 ABS complexes coated on the substrates. After 2h incubation, the concentration of the supernatants (red) reduced slightly for group-1 ABS, stronger for group-2 and strongest for group-3 ABS but no significant change was observed for control IgG.

3. Original comment/question. "2. Line 204: "Thus, both reverse and non-reverse experiments consistently show that the original sera likely contain polyclonal ABS with different binding affinities to PF4/H complexes and indicate that the capacity of anti-PF4/P-ABS to activate platelets depends on their binding strength to PF4/H complexes."

The authors' data convincingly show that Group-3 ABS recognize PF4/P complexes with higher affinity than Group-2 or 1 ABS and that the former ABS activate normal platelets even in the absence of low dose heparin. These findings are important, but it doesn't necessarily follow that the correlations reflect a cause-and-effect relationship. The authors' conclusions would be strengthened if they could identify what is being recognized on the platelet surface by Group-3 ABS in the absence of heparin and by the Group-2 ABS in the presence of heparin in the HIPA studies (Fig. 2). This question is very important. Surely it at least deserves some discussion."

New comment: In their response, the authors appear to be assuming that treating platelets with chondroitinase ABC removes all the chondroitin sulfate. In fact, such treatment removes less than half of the CS and about 24 hours is required to achieve that. Even if their treatment did remove all the CS, the findings described don't shed light on what the platelet target for Group 3 antibodies might be. And it is hard to imagine that meaningful atomic force studies can be done on platelets subjected first to chondroitinase treatment. It is not surprising that polyphosphates bind to PF4 but that seems irrelevant to the issue at hand.

Response: The Reviewer asks what group-3 ABS recognize on the platelet surface. We have shown that group-3 ABS bind PF4 in its native state by EIA, ITC, SMFS, DLS and a PF4 column and there is no reason to assume that they recognize anything else but PF4 on the platelet surface. Group-3 ABS also bind to PF4/heparin complexes as shown by EIA and we cannot differentiate for certain in which conformation PF4 is present on the platelet surface. We are aware of the risk of artifacts when platelets are treated with enzymes. We fully agree with the Reviewer that it does not make sense to treat platelets with chondroitinase for >24h before performing AFM experiments as we will create uncontrollable artifacts. We performed the additional experiment with chondroitinase to address the original comment of the Reviewer, but we are happy to remove now the chondroitinase experiments based on the technical suggestion of the Reviewer. Identification of the binding site of PF4 on the platelet surface needs to be addressed in further studies which are beyond the scope of the present manuscript.

4. Original comment/question Line 292: "Previously, we have shown that polyanions of

certain length and charge density induce the conformational change in PF4 (5, 28), required for binding of anti-PF4/P-ABS, by forcing two PF4 together (16). This results in fusion of the positive charge cloud of two PF4 tetramers and the release of additional energy, required to switch the structure of the PF4 protein for expression of the neo-epitope (16).

References 5 and 28 do show that various polyanionic compounds induce structural changes in PF4 but did not directly demonstrate any connection between these changes and HIT antibody binding. In Ref 16, it was shown that PF4 molecules repel each other because of their strong positive charges but that when PF4 is treated with glutaraldehyde to cross-link PF4 monomers "internally," weak PF4-PF4 interaction can be demonstrated by SMFS. The authors suggest this finding supports the view that PF4-PF4 bonds exist in clusters of PF4 aggregated by "long" heparin molecules. In the view of this reviewer, none of these data support the authors' contention (expressed in lines 292-296 and in many other places in the text) that Group-3 HIT ABS can induce changes in PF4 that lead to "...expression of the neoantigen" recognized by Group-1 and Group-2 ABS and in this sense can "substitute for heparin." The SMFS studies are consistent with this possibility but it is a real stretch to suggest that these findings can be used to draw conclusions about what happens in a patient.

New comment: The new data are helpful, but there is still a problem justifying the claim that the HIT "antigen" is created when 2 PF4 molecules are "pulled together." How does this gibe with the idea that polyanion-induced conformational changes in PF4 create the target recognized by HIT abs? Surely the two processes can't be structurally identical, or even similar. How do the authors reconcile this apparent conflict? Note that reviewer 2 also had a problem with the suggestion that Group 3 antibodies can "substitute" for heparin. "Substitute" just isn't the right word to use here. Why not describe the mechanics of this phenomenon and let it go at that?

Response: We agree that the two processes: a) polyanions bind to PF4 inducing a conformational change; and b) that group-3 ABS bind to PF4 inducing a conformational change cannot be structurally identical, as antibodies and linear polyanions are different molecules. What we want to describe is that in both cases, however, an epitope is exposed on PF4 which allows binding of PF4/polyanion specific antibodies. In immunohematology, the epitopes on molecules are typically defined by antibody binding. An antibody specific for a distinct epitope can only bind if the epitope is present. By showing binding of PF4/H specific group-2 ABS to PF4/group-3 ABS complexes, we directly prove that group-3 ABS induce a neoepitope on PF4, which is recognized by group-2 ABS. In the re-revised version of the manuscript, we removed all statements that ABS substitute for heparin.

5. Original comment/question Supplementary Fig 1: It is unusual to be able to elute antigen-specific IgG from serum in quantities sufficient to produce the bands shown in Suppl Fig 1. Were any studies done to determine experimentally what fraction of IgG present in the bands shown in lanes 9-11 is actually antibody specific for PF4/P? The gels could, for example, be performed before and after absorbing the eluates with beads coated with PF4/H complexes.

New comment: This response offered doesn't address the question, which was "What % of the IgG in the bands shown was actually HIT antibody?"

Response: We apologize for this unclear explanation. As the bands were all obtained using affinity purified PF4/H ABS, the vast majority of the IgG in the bands are HIT antibodies. We cannot absolutely exclude some contaminants during affinity purification of antibodies. For example, some non-PF4/H ABS may have bound to the column and are then released together with PF4/H ABS while eluting.

Fig. for Reviewers only. Purified PF4/H ABS were retested by absorbing on PF4/H coated beads. Only less than 5% of ABS are non-PF4/H ABS.

To check if our purified ABS are 100% PF4/H ABS, we now test the purified ABS again by incubating them with PF4/H coated beads and determine the concentration of the non-PF4/H ABS in the supernatant. The results show that >95% ABS in the eluates from PF4/H column again bind to PF4/H coated beads or >95% purified ABS are PF4/H ABS.

6. Original comment/question KKO is supposed to be a potent activator of platelets, yet it behaved like a Group-1 antibody in these studies. Some comment about this unexpected finding would be appropriate.

New comment: In their response, the authors seem to be suggesting that KKO may have a lower off rate yet also a lower affinity. The authors need to be careful about this argument. By “smaller” k_{off} , do they mean a smaller off rate? Antibody affinity is strictly governed by the off rate. If an antibody persists longer on its target, it will have a smaller off rate and a higher affinity. So the explanation offered for the behavior of KKO seems untenable.

Response: We fully agree with the Reviewer that ‘lower off-rate’ means higher binding affinity as we have always stated in the result and discussion part that the lower k_{off} = higher binding affinity. Here is the paragraph in the Results part: “We found that group-1 ABS have a slightly lower binding affinity ($k_{off} = 15.6 \text{ s}^{-1}$) than group-2 ABS ($k_{off} = 2.0 \text{ s}^{-1}$), or KKO ($k_{off} = 2.2 \text{ s}^{-1}$), while group-3 ABS had the highest binding affinity ($k_{off} = 0.12 \text{ s}^{-1}$) (Supplementary Fig. 5). These results indicate that complexes formed by PF4/H complexes with group-3 ABS are more stable than those with group-1 and group-2 ABS, or with KKO.”

In the Discussion part, we stated “Interestingly, the thermal off-rates (k_{off}) are similar for KKO ($k_{off} = 2.2 \text{ s}^{-1}$) and group-2 ABS ($k_{off} = 2.0 \text{ s}^{-1}$), while it is higher for group-1 ABS ($k_{off} = 15.6 \text{ s}^{-1}$). The low off-rate likely allows long enough binding of the antibodies to platelets to induce platelet activation by cross-linking $\text{Fc}\gamma\text{RIIA}$.”

Reviewer #2 (Remarks to the Author):

The authors gave a thorough response, including numerous additional experiments that indeed did strengthen their proposal on the mechanism of action of HIT antibodies.

The ultimate proof that group 3 ABS are revealing the neo-epitope allowing group 2 ABS to bind and activate platelets, and thus truly mimic heparin/polyanion in this respect, indeed would be to use F(ab)'_2 -fragments of group 3 ABS in combination with group 2 intact ABS.

This is mentioned by the authors under "Outlook" to be impossible "as the polyclonal patient serum also contains weaker reacting, polyanion-dependent (group-2) PF4/P-ABS. Their F(ab)'_2 fragments would block binding of intact polyanion-dependent anti-PF4/P-ABS."

Probably correct, however if F(ab)'_2 fragments would be made from PF4-purified group 3 ABS, the relative low number of remaining group 2 F(ab)'_2 likely would no longer be a problem.

Response: This is a nice suggestion from the Reviewer. We do not have enough human sera containing group-3 ABS in order to carry out these experiments. Production of F(ab)'2 and their purification makes sense if we have at least 1-2 mg of purified antibodies. We have added in the "Outlook" this suggestion of using F(ab)'2 fragments.

Reviewer #3 (Remarks to the Author):

The manuscript has significantly improved, yet the force spectroscopy part (Fig. S6) is still completely unclear. Which forces from the broad force distributions have been taken for Fig. S6. Indicate widths of distributions in Fig.S6, error bars, quality of fits etc.

Response: We thank the Reviewer for pointing out this unclear description. With pre-tested cantilevers coated with group-2 or group-3 ABS to select the strong reactivity ABS, we did not measure wide distributions of binding forces when these preselected antibodies were used. The mean rupture force from each experiment was determined by applying Gaussian fits as described in the experimental section. Then, the mean rupture force at each loading rate measured from three sera per antibody groups was averaged to obtain a median together with a standard deviation (shown in Fig.S5). We have now added this way of analysis to the 'Data analysis'.

REVIEWERS' COMMENTS:

Reviewer #3 (Remarks to the Author):

The authors did respond in a satisfactory and/or convincing manner to the questions raised. I therefore do not have further remarks.

Reviewer #4 (Remarks to the Author):

My questions have been answered.